# Completeness and Coherence Learning for Fast Arbitrary Style Transfer

**Zhijie Wu**                                                          *zhijiewu@cs.ubc.ca*
*Department of Computer Science*
*University of British Columbia*

**Chunjin Song**                                                       *chunjins@cs.ubc.ca*
*Department of Computer Science*
*University of British Columbia*

**Guanxiong Chen**                                                     *gxchen@cs.ubc.ca*
*Department of Computer Science*
*University of British Columbia*

**Sheng Guo** ✉                                                        *guosheng1001@gmail.com*
*MYbank, Ant Group*

**Weilin Huang**                                                       *weilin_h@hotmail.com*
*Alibaba Group*

**Reviewed on OpenReview:** *https://openreview.net/forum?id=4N6T6Rop6k*

## Abstract

Style transfer methods put a premium on two objectives: (1) *completeness* which encourages the encoding of a complete set of style patterns; (2) *coherence* which discourages the production of spurious artifacts not found in input styles. While existing methods pursue the two objectives either partially or implicitly, we present the Completeness and Coherence Network (CCNet) which jointly learns completeness and coherence components and rejects their incompatibility, both in an explicit manner. Specifically, we develop an attention mechanism integrated with bi-directional softmax operations for explicit imposition of the two objectives and for their collaborative modelling. We also propose CCLoss as a quantitative measure for evaluating the quality of a stylized image in terms of completeness and coherence. Through an empirical evaluation, we demonstrate that compared with existing methods, our method strikes a better tradeoff between computation costs, generalization ability and stylization quality.

1

## 1 Introduction

Regarding neural style transfer problems (Jing et al., 2020b), Gatys et al. (2016) has demonstrated that a pretrained VGG network (Simonyan & Zisserman, 2015) can produce features that entail content structures and style patterns in their correlation. But the optimization-based method (Gatys et al., 2016) is prohibitively slow and its range of application is fairly limited. Since then a great number of efforts (Johnson et al., 2016; Chen et al., 2017; Sanakoyeu et al., 2018; Gu et al., 2018; Chen et al., 2021; Deng et al., 2022) have been dedicated to striking the balance between speed, generalization capability and stylization quality. To achieve fast and arbitrary style transfer, Statistics-based methods adjust the holistic statistics of a content image

---

[1]✉: Corresponding author

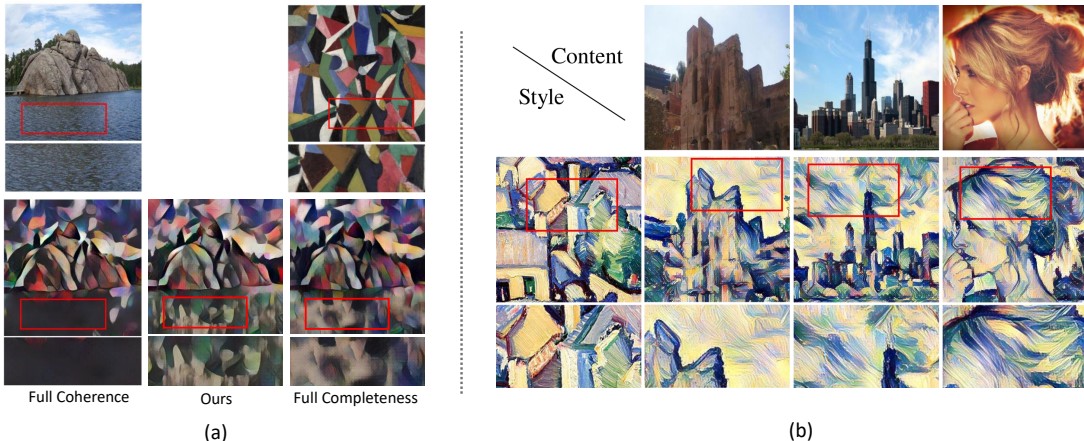

Figure 1: (a) Given a content-style image pair, the full coherence learning of input styles produces concrete but incomplete patterns while the full completeness learning presents rich textures with white distorted artifacts. (b) Instead, CCNet can capture notable holistic styles comprehensively and fine-grained details faithfully by jointly capturing completeness and coherence.

in accordance to that of a style image (Huang & Belongie, 2017; Li et al., 2017; Sheng et al., 2018; Zhang et al., 2019; Li et al., 2019; Wu et al., 2020). These methods can capture a diverse set of styles (indicating *completeness*) but often fail to synthesize style characteristics faithfully, as they often introduce unexpected or distorted patterns. Patch-based methods on the other hand swaps local feature patches (Chen & Schmidt, 2016; Park & Lee, 2019). Although these methods do not suffer much from pattern distortions, and excel at synthesizing fine-grained style details (indicating *coherence*), they often cannot synthesize a complete set of input styles, and they repeat undesirable style patterns such that blurred contents are produced.

The aforementioned two lines of work dedicates to either completeness or coherence. Others have sought to achieve both objectives at once: Gu et al. (2018); Sheng et al. (2018); Park & Lee (2019); Wu et al. (2021); Liu et al. (2021); Deng et al. (2022); Zhang et al. (2022) have already attempted to unify statistics-based and patch-based methods. Yet their solutions are either partial or indirect: without an explicit, direct formulation of coherence or completeness, fully exploiting the two transfer priors is impossible. Seeing the shortcomings of our predecessors, we formally define completeness and coherence in the context of style transfer: for an input style feature $F_s$ and initial stylized result $F_{cs}$, completeness requires as many patches in $F_s$ as possible to be preserved in $F_{cs}$; coherence requires all patches in $F_{cs}$ to be from $F_s$ (Eq. 1). In fact, our formulation of the two transfer priors resonate with symmetric Chamfer matching (Simakov et al., 2008; Fan et al., 2017; Yang et al., 2018). Our explicit definition allows us to 1) explicitly impose completeness and coherence as stylization objectives that lead to unbiased results; 2) parse the relation between completeness and coherence features for compatibility learning; 3) identify the optimal trade-off between the two objectives by tuning their relative weights. To this end, we introduce a patch-based similarity measure (Simakov et al., 2008) to both forward propagation during inference and back propagation during training.

Hence we devise the Completeness and Coherence Network (CCNet), a dual-branch style transfer framework with feature diffusion networks inside to leverage our formulation of completeness and coherence. The gist of the CCNet is the Non-local Diffusive Attention Module, which further stylizes $F_{cs}$ by completeness and coherence while preserving its spatial structures: based on the semantic correlation between $F_{cs}$ and $F_s$ (Sheng et al., 2018), we update each patch in $F_{cs}$ by its closest patch in $F_s$ for coherence modeling. Similarly, we propagate each patch in $F_s$ to its closest patch in $F_{cs}$ for completeness modeling. This joint analysis paradigm aims to learn the interrelation of these objectives, as we assume that compatibility between completeness and coherence feature leads to better stylization.

Turning to our implementation—lying at the heart of the Non-local Diffusive Attention Module is a bidirectional softmax operator (Fig. 5): each direction of the operator is situated in one branch, for CCNet to learn one of the two objectives at low computational cost, before the two branches collaborate for joint

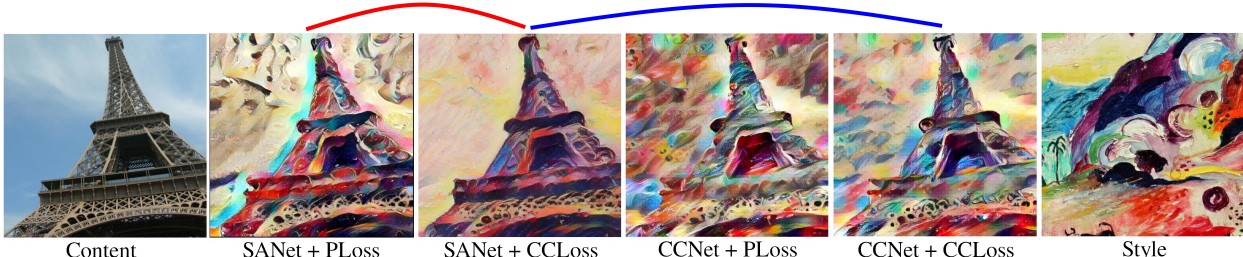

| Content | SANet + PLoss | SANet + CCLoss | CCNet + PLoss | CCNet + CCLoss | Style |

Figure 2: We show that both CCNet (for feed-forward inference) and CCLoss (for backward training) are effective by comparing them with SANet and perceptual loss. Compared with perceptual loss, CCLoss significantly reduces distorted patterns in the background and halation around edges (check the patches joined by the red curve). Compared with SANet, CCNet renders a more diverse set of styles, as we can see from the color variation in the sky (check the patches joined by the blue curve). Neither CCNet (for forward propagation) nor CCLoss (for back-propagation) can lead to satisfactory performance on its own. Check the appendix for more results.

analysis. Finally, a learnable decoder reconstructs a stylized image from learned features. Inspired by prior work in which symmetric Chamfer matching was used to formulate losses (Simakov et al., 2008; Fan et al., 2017; Yang et al., 2018), we propose the completeness and coherence loss (CCLoss) as a patch-based similarity measure to train our network. Unlike the holistic perceptual loss (PLoss) (Johnson et al., 2016), CCLoss allows us to explicitly balance between completeness and coherence. Its patch-wise nature also helps with representing detailed style patterns (Fig. 1). We also apply the identity loss (Park & Lee, 2019) to speed up training and to maintain content structure without compromising style richness.

To the best of our knowledge, we are the first to explicitly capture completeness and coherence and analyze them altogether in a patch-based manner. Our contributions are as follow:

- We explicitly introduce completeness and coherence to style transfer, and we present a novel framework (CCNet) and a loss function (CCLoss) to fully exploit the two objectives via joint analysis.
- We design an improved softmax operator (Fig. 5) for fast completeness and coherence modeling, and we investigate the impacts of our architectural design choices via a comprehensive ablation study.
- We demonstrate that CCNet is better at capturing diverse and coherent style patterns than existing methods, and that it achieves better stylization quality without incurring heavey computation burdens.

## 2 Related Work

**Style transfer.** Here we only review the most relevant methods for neural arbitrary style transfer, and we refer readers to Jing et al. (2020b) for a comprehensive survey.

Gatys et al. (2016) is a pioneer work that achieved impressive stylization results. Nevertheless it was built upon a time-consuming optimization method; moreover, it attends only to completeness, but does not address style pattern distortions at all (Li & Wand, 2016a; Chen & Schmidt, 2016; Sheng et al., 2018). Since then, Chen & Schmidt (2016) realized fast arbitrary style transfer with a patch-swap operation for intermediate learned features, but the method cannot parse the complete set of style information. Several later approaches (Huang & Belongie, 2017; Li et al., 2017; Sheng et al., 2018; Song et al., 2019; Li et al., 2019; Jing et al., 2020a) propose to replace the statistics of content images with those of style images. Taking advantage of the strong representation power of second-order statistics, they (Li et al., 2017; Song et al., 2019) are able to render a rich set of style elements, hence achieving completeness. But these methods often produce spurious artifacts and distort both spatial layouts and style patterns in their renderings, thus being incoherent with respect to their inputs. SANet (Park & Lee, 2019) can produce the most coherent results w.r.t. input styles, since it directly combines content features with their closest patches from style features. But it often biases too much towards contents because the matched style patches strictly follow local content structures. Therefore it often fails to render a complete representation of input styles.

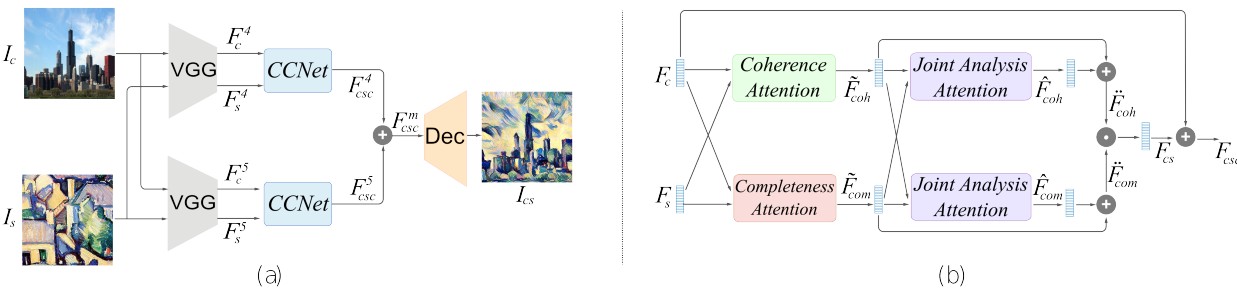

Figure 3: (a) Architecture of the entire style transfer pipeline. A VGG-based encoder transforms content image $I_c$ and source image $I_s$ to $F_c^i$ and $F_s^i$ for $i \in \{4, 5\}$ respectively. Each feature map is then fused by a CCNet module. The output from each CCNet is then summed up and passed to a trainable decoder to produce images $I_{cs}$. (b) Architecture of a CCNet module. Content feature $F_c$ and source feature $F_s$ are passed to diffusive attention modules (Fig. 4) to model completeness and coherence. Then $\tilde{F}_{coh}$ and $\tilde{F}_{com}$ are passed to Joint-Analysis Attention Module to compute skip connections for better compatibility between $\tilde{F}_{coh}$ and $\tilde{F}_{com}$. Finally, the stylization feature $F_{cs}$ is produced via an element-wise multiplication.

While the aforementioned work can achieve either completeness or coherence but not both, later work (Gu et al., 2018; Sheng et al., 2018; Liu et al., 2021; Deng et al., 2022) aim to model both completeness and coherence, but only indirectly without individually formulating completeness and coherence features. Specifically, Gu et al. (2018) proposes a feature reshuffle module to boost style richness by reducing the chances to search similar style patches. Sheng et al. (2018) matches normalized feature maps before adjusting their statistics to mitigate conflicts between content structures and target styles. Liu et al. (2021) boosts its ability to synthesize details by adaptively performing attentive normalization on a per-point basis. Other similar state-of-the-art methods include internal-external contrastive learning (Chen et al., 2021), parametric style composition (Wu et al., 2021), adversarial learning (Xu et al., 2021), vision transformer (Deng et al., 2022) and domain enhancement based style projector (Zhang et al., 2022).

For arbitrary stylization, our approach follows the local patch alignment paradigm, which is closely related to patch-based methods (Li & Wand, 2016b; Chen & Schmidt, 2016; Sheng et al., 2018). But unlike these methods, we align images by explicitly imposing the completeness and coherence of input styles and jointly analyzing them. This transfer strategy can simultaneously preserve the richness and fine-grained details of style patterns better than prior. While other methods have yet to achieve this, the CCLoss allows us to balance completeness and coherence by adjusting their relative weights. As an ablated form of CCNet, SANet (Park & Lee, 2019) only takes coherence into account; see Fig. 2 and Sec. 3.2 for detailed discussions.

**Symmetric Chamfer matching.** Chamfer matching (Barrow et al., 1977) has a multitude of applications in computer vision and graphics (Borgefors, 1988; Thayananthan et al., 2003; Ma et al., 2010; Wu et al., 2019). Simakov et al. (2008) first introduced symmetric Chamfer matching to the field of visual summary by explicitly defining completeness and coherence. In the context of deep learning, several work (Fan et al., 2017; Achlioptas et al., 2018; Yang et al., 2018) have applied similar ideas to reconstruct 3D point clouds. Since then, symmetric Chamfer distance has been extensively adapted to measure the synthesized quality of 3D shapes. But to the best of our knowledge, we are the first to realize style transfer with explicit and full completeness and coherence modeling. Unlike existing methods, we perform completeness and coherence modeling in both feed-forward style inference and backward network training to maximize their representation potentials. Moreover, we aim to jointly analyze these two objectives at inference. We present an evaluation of these design components in Fig. 2, Fig. 8 and in the ablation study in Sec. 4.

## 3 Method

Fig. 3 illustrates the overall architecture of our style transfer pipeline. Our discussion follows a bottom-up approach: we start off with detailing the gist of our CCNet—the Non-local Diffusive Attention Module in Sec. 3.1. The module leverages affinity matrix (Jiang et al., 2018; Wang et al., 2018; Zhu et al., 2019) which allows the output style feature to encapsulate richer style patterns and more fine-grained details. Then in Sec. 3.2

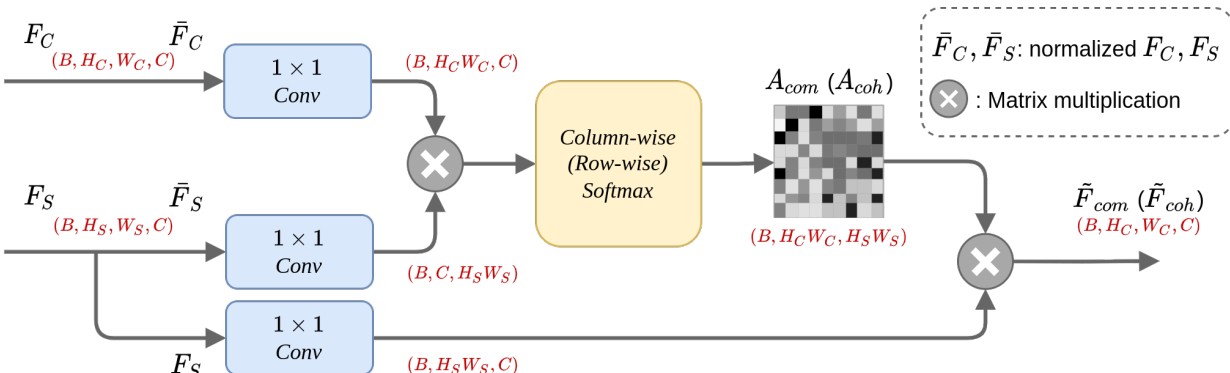

Figure 4: The Non-local Diffusive Attention Module as a fundamental building block of the CCNet in Fig. 3 (b). Given $F_c$ with shape $(B, H_c, W_c, C)$ and $F_s$ with shape $(B, H_s, W_s, C)$ as input content & style features, we use a diffusion network to produce completeness (coherence) feature $\tilde{F}_{com}$ ($\tilde{F}_{coh}$).

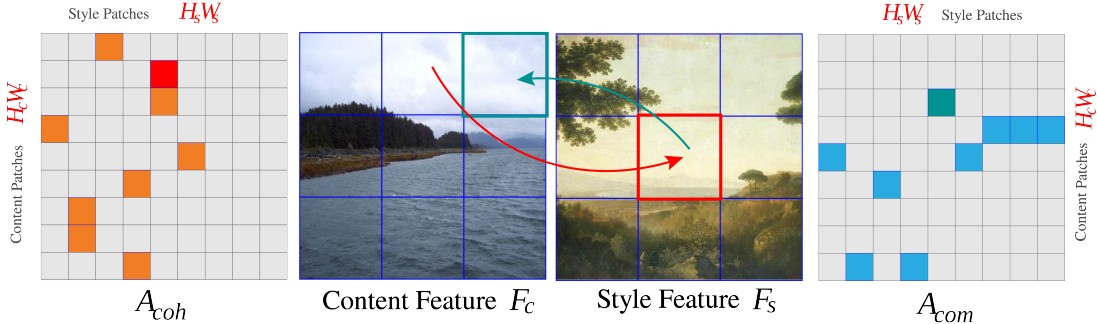

Figure 5: Here we detail the five steps to compute $\tilde{F}_{com}$ from Fig. 4: 1) First, we divide content feature $F_c$ and source feature $F_s$ into 9 patches. 2) Then we compute a $9 \times 9$ affinity matrix $A$ to measure affinity between content patches from $F_c$ and style patches from $F_s$. 3) Next, we apply the softmax function along each column of A to produce the completeness matrix $A_{com}$. Each of its element $A_{com}^{i,j}$ tells how much style information from the $j$-th style patch should propagate to the $i$-th content patch. 4) So for each style patch, we can identify its "nearest content patch" with the largest value $\max_i A_{com}^{i,j}$ (see the blue cells) to ensure every style patch can be included in the stylized content feature, hence enriching stylized patterns. 5) Finally, for completeness transfer, we diffuse each source style patch to its closest content patch through $\tilde{F}_{com} = A_{com} \cdot F_s$ to encourage more diverse results. Here "·" is a dot product. Similarly, we can compute the coherence matrix ($A_{coh}$) by computing softmax along each row of $A$, identify the largest values (orange cells), look up for the nearest style patch for each content patch and finally produce $\tilde{F}_{coh}$. We visualize the closest content patch (w.r.t) a style patch in teal and a closest style patch in red.

we discuss the Completeness-Coherence Network (CCNet) module with four attention modules instantiated inside. While the Coherence (Completeness) Attention Module focuses solely on its own objective, each Joint Analysis Attention Module processes the learned coherence and completeness feature collaboratively. In Sec. 3.3 we discuss our overall architecture, with emphasis on the CCLoss which is for balancing the effects of completeness and coherence while preserving details. Finally we discuss implementation details in Sec. 3.4. For more discussions on our model design, please refer to the appendix.

## 3.1 The Non-local Diffusive Attention Module

Inspired by Fan et al. (2017); Simakov et al. (2008), we assume a stylized feature $F_{cs}$ is visually coherent and complete to style feature $F_s$ if as many as possible patches of $F_{cs}$ are preserved in $F_s$, and vice versa. Namely, for each stylized content patch in $F_{cs}$, we search for its most similar patch in $F_s$ and evaluate their

distance, and vice-versa:

$$d(F_{cs}, F_s) = \sum_{P \subset F_{cs}} \min_{Q \subset F_s} D(P, Q) + \sum_{Q \subset F_s} \min_{P \subset F_{cs}} D(Q, P), \tag{1}$$

where P and Q denote patches in $F_{cs}$ and $F_s$ respectively. The first term in Eq. 1 measures deviation from coherence; the second term measures deviation from completeness. Neither completeness nor coherence on its own is enough to produce a good style transfer.

We use $F_c$ to denote initial stylized feature, and we use it alongside $F_s$ to compute final stylized feature $F_{cs}$. Similar to Simakov et al. (2008), we propagate $F_s$ to $F_c$ in patches to optimize the similarity in Eq. 1. Specifically, for each patch $Q \subset F_s$, we find its closest patch $P \subset F_c$ and apply the patch $Q$ to update $P$ such that all style patches in $F_s$ are incorporated into $F_{cs}$, leading to more diverse stylized results. Looking at the other way round: for each patch $\dot{P} \subset F_c$, we find the closest patch $\dot{Q} \subset F_s$ and apply the patch $\dot{Q}$ to update $\dot{P}$ for coherence, leading to more adaptive results. We then implement this symmetric nearest neighbor search with a softmax operation along each axis of an affinity matrix (check Fig. 5 for details). This feature update procedure encourages us to employ the non-local diffusion architecture (Jiang et al., 2018; Wang et al., 2018) in Fig. 4 to achieve this motivation. We also create instances of the non-local diffusive module to jointly analyze completeness and coherence for compatibility learning. Although overlapping patches with larger sizes are better at capturing coarse scales and modeling relationships within a local region, we set the patch size to 1 in order to balance visual performance and computation cost.

### 3.2 The Completeness and Coherence Network (CCNet)

CCNet (Fig. 3) computes the stylized feature $F_{cs}$ by spatially rearranging $F_s$ with $F_c$. To do this, we create multiple instances of the Non-local Diffusive Attention Module: *Coherence* and *Completeness Attention* for modelling completeness and coherence, followed by two *Joint Analysis Attention* modules.

Following Sheng et al. (2018); Park & Lee (2019), we first normalize $F_c$ and $F_s$ to remove their style information so that the later style diffusion can be realized based on the content structures of input images, yielding $\bar{F}_c$ and $\bar{F}_s$. Turning out attention to the Completeness (Coherence) Attention Module: Let $softmax_i(\cdot)$ be a softmax operation along the $i^{th}$ axis and the starting index of axes be 0. Following our discussion in Sec. 3.1, we compute features for coherence and completeness as[2]:

$$\tilde{F}_{coh} = softmax_2(\psi_u^{coh}(\bar{F}_c) \cdot \psi_g^{coh}(\bar{F}_s)^T) \cdot \psi_h^{coh}(F_s), \tag{2}$$

$$\tilde{F}_{com} = softmax_1(\psi_u^{com}(\bar{F}_c) \cdot \psi_g^{com}(\bar{F}_s)^T) \cdot \psi_h^{com}(F_s). \tag{3}$$

Here $\{\psi_h^{coh}, \psi_u^{coh}, \psi_g^{coh}, \psi_h^{com}, \psi_u^{com}, \psi_g^{com}\}$ and "·" represent learnable convolutions and dot-product similarity individually; see Fig. 5 for more details.

Next, we feed $\tilde{F}_{coh}$ and $\tilde{F}_{com}$ into the Joint Analysis Attention Module (Fig. 3 (b)). We instantiate Joint Analysis Attention for two reasons: 1) We aim to improve pixel-level compatibility between $\tilde{F}_{coh}$ and $\tilde{F}_{com}$; 2) We aim to combine the information at different positions of an image to capture the long-range dependencies between pixels. To these ends, we feed $\tilde{F}_{coh}$ and $\tilde{F}_{com}$ to another shared diffusion block which allows us to rearrange feature vectors of $\tilde{F}_{coh}$ ($\tilde{F}_{com}$) to fit $\tilde{F}_{com}$ ($\tilde{F}_{coh}$) well; see the appendix for more details. Thus we compute two residual features as:

$$\hat{F}_{coh} = softmax_2(\psi_u(\tilde{F}_{com}) \cdot \psi_g(\tilde{F}_{coh})^T) \cdot \psi_h(\tilde{F}_{coh}), \tag{4}$$

$$\hat{F}_{com} = softmax_2(\psi_u(\tilde{F}_{coh}) \cdot \psi_g(\tilde{F}_{com})^T) \cdot \psi_h(\tilde{F}_{com}), \tag{5}$$

where $\{\psi_h, \psi_u, \psi_g\}$ denotes learnable convolution parameters and "·" denotes dot-product similarity. Then we update $\tilde{F}_{coh}$ and $\tilde{F}_{com}$ as $\ddot{F}_{coh} = \tilde{F}_{coh} + \hat{F}_{coh}, \ddot{F}_{com} = \tilde{F}_{com} + \hat{F}_{com}$. And we fuse $\ddot{F}_{coh}$ and $\ddot{F}_{com}$ together as $F_{cs} = \ddot{F}_{coh} \odot \ddot{F}_{com}$ to further facilitate their compatibility in a channel-wise manner, where $\odot$ indicates an element-wise multiplication. Finally, we merge $F_c$ into $F_{cs}$ to better preserve the input content structures as $F_{csc} = F_{cs} + F_c$.

---

[2]Both $\psi_u^{coh}(\bar{F}_c) \cdot \psi_g^{coh}(\bar{F}_s)^T$ and $\psi_u^{com}(\bar{F}_c) \cdot \psi_g^{com}(\bar{F}_s)^T$ measure the affinities between $\bar{F}_c$ and $\bar{F}_s$ whose shape is [batch size, resolution of $F_c$, resolution of $F_s$]; check details in the appendix.

**Discussion on SANet.** Both our CCNet and SANet use the non-local network (Wang et al., 2018) as a fundamental building block, since both of our networks measure feature correlations and then diffuse target style features to content features. But our approach differs from that of SANet in the follow aspects:

- As mentioned in Sec. 2, our motivation differs from SANet's. Unlike SANet which matches semantically nearest style features onto content features, we take complete styles into account. Our approach allows us to address coherence and completeness more comprehensively than existing patch-based and statistics-based methods, as shown in Fig. 2 and 6.
- As far as model architecture, we can think of SANet as an ablated form of CCNet: if we keep the Coherence Attention Module only, and strip away the Completeness Attention Module as well as the Joint Module then we have SANet. In Fig. 2 and 8, we show that CCNet with minor architectural changes from SANet yields remarkable visual improvement.
- SANet is trained with perceptual loss (Johnson et al., 2016), which generates style distortions and introduces blurry halation around the edges in Fig. 2. Our CCLoss yields much neater contours and less distorted style patterns via coherence modeling and patch-wise computation.

### 3.3 Style Transfer Pipeline with CCNets

Aside from putting in the CCNet modules, in our style transfer pipeline we retain the most of SANet's architecture, so that when benchmarking for completeness and coherence we can attribute any performance improvement directly to CCNet: see Fig. 2 and Fig. 6. In Fig. 3 (a), our entire pipeline consists of three parts: one VGG-based encoder (denoted as $E$), a symmetric decoder and two CCNet modules. The VGG-based encoder takes a content image $I_c$ and a style image $I_s$ as inputs, and produces content feature map $F_c^i$ and style feature map $F_s^i$, where $i$ stands for the map being produced by the $relu\_i$ layer of the encoder. We instantiate two CCNet modules in our framework to capture the style patterns at multiple scales with modest computational cost. Each CCNet takes the content and style feature map from a single layer ($relu\_4$ or $relu\_5$) as inputs and synthesizes the stylized features as:

$$F_{csc}^4 = CCNet(F_c^4, F_s^4), \ F_{csc}^5 = CCNet(F_c^5, F_s^5). \tag{6}$$

Then $F_{csc}^4$ and $F_{csc}^5$ are fused as:

$$F_{csc}^m = F_{csc}^4 + u(F_{csc}^5), \tag{7}$$

where the $u(\cdot)$ stands for upsampling. Finally, a trained decoder maps $F_{csc}^m$ back to a stylized image $I_{cs}$ which possesses the content structures from $I_c$ and the style patterns from $I_s$.

**Loss function.** Our loss function for jointly training the two CCNets and the decoder consists of two parts:

$$L_{total} = L_{id} + L_{cc}. \tag{8}$$

$L_{id}$ is the identity loss (Park & Lee, 2019) which speeds up training and maintains the content structure without losing richness of styles. Following Park & Lee (2019), we define the identity loss as $L_{id} = \lambda_{id}^1 L_{id}^1 + \lambda_{id}^2 L_{id}^2$, and both $L_{id}^1$ and $L_{id}^2$ are computed as:

$$L_{id}^1 = ||I_{cc} - I_c||_2 + ||I_{ss} - I_s||_2, \tag{9}$$

$$L_{id}^2 = \sum_{i=1}^{L} ||E^i(I_{cc}) - E^i(I_c)||_2 + ||E^i(I_{ss}) - E^i(I_s)||_2, \tag{10}$$

where $I_{cc}$ (or $I_{ss}$) denotes reconstruction results from stylizing two identical content images (or style images) and $E^i(\cdot)$ indicates the features outputted by the layer $relu\_i$ of the VGG encoder.

$L_{cc}$ is the Completeness and Coherence Loss (CCLoss), which we use to facilitate the fine style patterns and control relative effects of completeness and coherence. Following Eq. 1 we set patch size to 1 and employ the

cosine distance as our affinity measure. Then we have $L_{cc} = \lambda_{cc}^{com} L_{cc}^{com} + \lambda_{cc}^{coh} L_{cc}^{coh}$ as:

$$L_{cc}^{coh} = \sum_{p^5 \in E^5(I_{cs})} \min_{q^5 \in E^5(I_s)} D(p^5, q^5) + \sum_{p^4 \in E^4(I_{cs})} \min_{q^4 \in E^4(I_s)} D(p^4, q^4), \tag{11}$$

$$L_{cc}^{com} = \sum_{q^5 \in E^5(I_s)} \min_{p^5 \in E^5(I_{cs})} D(q^5, p^5) + \sum_{q^4 \in E^4(I_s)} \min_{p^4 \in E^4(I_{cs})} D(q^4, p^4), \tag{12}$$

where $\{p^i, q^i\}, i \in \{4, 5\}$ are achieved via the encoder $E^i(\cdot)$ for transfer results and input style images individually. The distance between $p^i$ and $q^i$ is:

$$D(p^i, q^i) = 1 - \frac{\langle p^i, q^i \rangle}{\|p^i\| \cdot \|q^i\|}. \tag{13}$$

Here we only calculate the CCLoss for feature maps produced by the $relu\_4$ and $relu\_5$ layer of the encoder, corresponding to the input to each CCNet.

### 3.4 Implementation Details

We use $80,000$ images from MS-COCO (Lin et al., 2014) and $80,000$ images from WikiArt (Nichol, 2016) as the content and style dataset respectively for training. We initialize the encoder with a pre-trained VGG network (Simonyan & Zisserman, 2015) and freeze it during training. As far as the decoder, we take the same setting from Huang & Belongie (2017). We also apply the Adam optimizer (Kingma & Ba, 2015) with batch size set to four image pairs, and learning rate set to 1e-4 for 200K iterations. During training, first we resize the smaller dimension of each image to 512 but keep the initial ratio. Then we randomly crop a region of size $256 \times 256$. But in testing an input image can be of any size. Throughout our experiments, we set $\lambda_{id}^1$, $\lambda_{id}^2$, $\lambda_{cc}^{com}$ and $\lambda_{cc}^{coh}$ respectively to 50, 1, 300 and 5.

## 4 Experimental Results

First we compare our approach with several state-of-the-art methods qualitatively and quantitatively. Note that all of our results on the baselines are obtained from publicly available, pre-trained models under their default settings. Then we show results from our ablation study in which we investigate the impact of several design decisions. At last, we showcase two runtime applications of our method to demonstrate the flexibility of our model. In the appendix we show a more lightweight version of the Non-local Diffusive Attention Module, and we provide more details on our experiments. We will release our source code upon publication.

**Qualitative comparison.** We present qualitative results from our method and the baselines in Fig. 6 and Fig. 7. While evaluating artistic style transfer is still an open problem within the vision community (Li et al., 2017; Zhang et al., 2022), we focus on matters the community deems to be the most pressing: style distortions (Sheng et al., 2018), clear outlines of prominent objects (Zhang et al., 2019), full style modeling (Li et al., 2017) and reduction of repetitive patterns (Park & Lee, 2019).

In Fig. 6, Gatys et al. (2016) achieves arbitrary style transfer with a slow optimization method and often generates unstable results with distorted spatial layouts and style decoration (e.g. $1^{st}$ and $4^{th}$ row). While AdaAttn (Liu et al., 2021) can preserve content structures well, it fails to capture some prominent colors (e.g. the conspicuous oranges in $4^{th}$ row) and target textures (e.g. the block-wise patterns with black outlines in the $2^{nd}$ row) in most samples. It only adjusts colors for some but not all regions and produces spurious blob-like artifacts in the background (e.g. the $1^{st}$ and $3^{rd}$ row). WCT (Li et al., 2017) enhances its style representation ability by matching the covariance matrices of style features. However, it cannot produce fine-grained styles (e.g. the plume-like textures in $3^{rd}$ row and block-wise appearances in the $5^{th}$ row) and often blurs the content structures with distorted patterns (e.g. the $2^{nd}$ row). AvatarNet (Sheng et al., 2018) introduces a feature decorator to facilitate the adaptive style patterns, but still blurs the fine ingredients (e.g. the plume-like textures in the $3^{rd}$ row and brush strokes in the $4^{th}$ and $6^{th}$ row). It can not keep semantic structures (e.g. the $2^{nd}$ and $4^{th}$ row) as well and its background is overlaid with unseen colors (e.g. the $3^{rd}$ row). For SANet (Park & Lee, 2019), its style feature alignment biases towards content structures

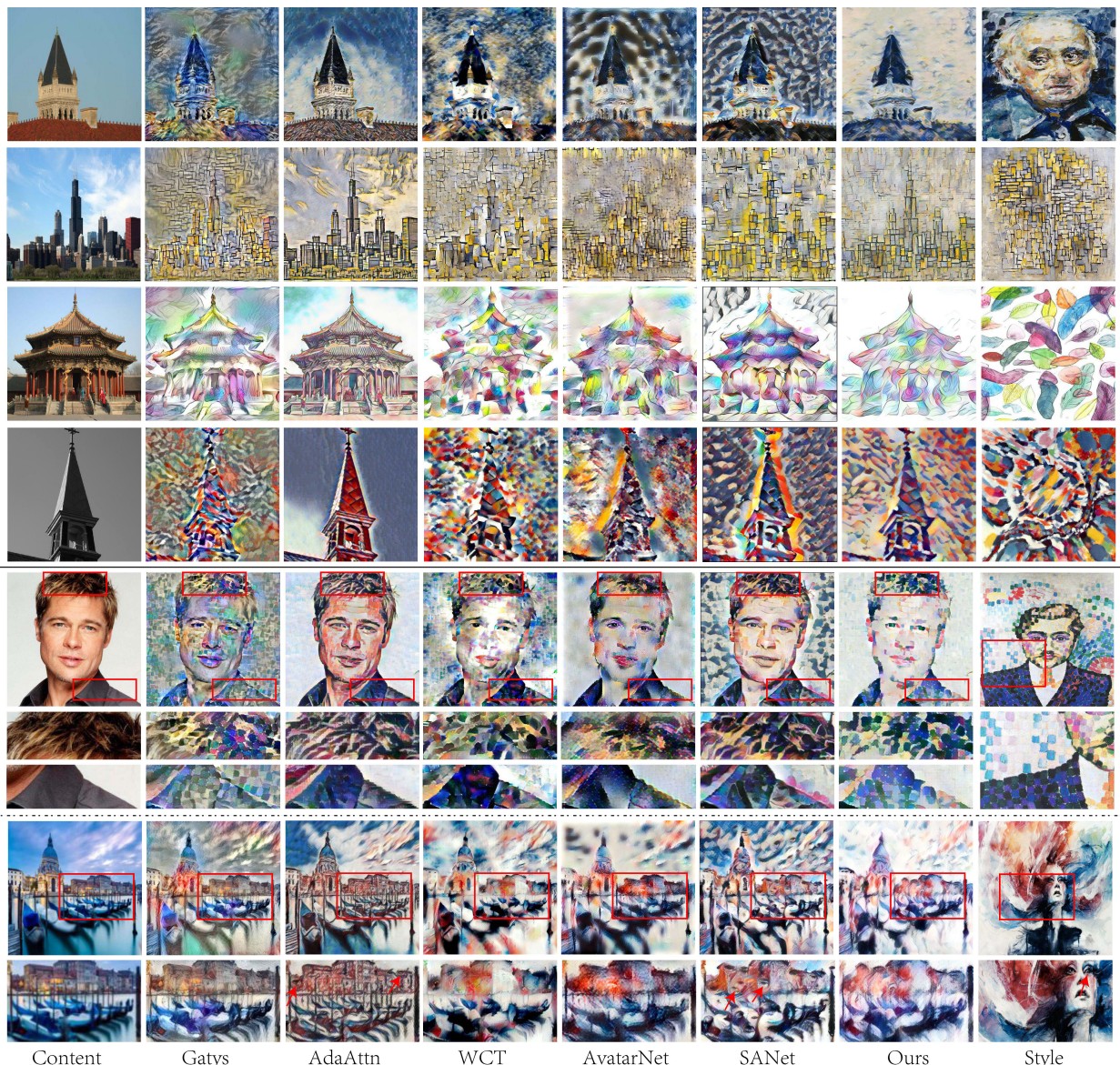

Figure 6: Comparison with prior methods. As mentioned in Sec. 4, we compare our method with the prior ones on how well we tackle the challenges that the style transfer community deems to be critical: style distortions (Sheng et al., 2018), blurry object outlines (Zhang et al., 2019; Park & Lee, 2019), repetitive patterns (Zhang et al., 2019) and lack of style richness (Li et al., 2017). Our method is better at reflecting full style elements (e.g. color distribution in the sky of $4^{th}$ row), introducing faithful textures (e.g. plume-like textures in $3^{rd}$ row), generating neat contours (e.g. $1^{st}$, $2^{nd}$, $3^{rd}$ and $4^{th}$ rows) and reducing distorted patterns in background (e.g. $1^{st}$ and $3^{rd}$ rows). Looking into visual details, we see CCNet preserves richer fine-grained textures (e.g. the block-wise appearances with different colors in $5^{th}$ row and the brushstroke-wise patterns in last row). Neither SANet nor AdaAttn can perceive fine-grained textures ($5^{th}$ row); they also repeat the eye pattern over the entire image (as red arrows shown in last row) which is not desirable. SANet also introduces blurry edges with halation and generates unseen clutters in background in the $5^{th}$ row. Also, AvatarNet and WCT misrepresent the input block-wise and brushstroke-wise style details and Gatys et al. (2016) fails to capture the holistic color distribution. More detailed evaluation is included in the appendix; zoom in for better visualizations.

too much. Therefore it cannot always parse a complete set of style patterns (e.g. colors in the sky of $4^{th}$

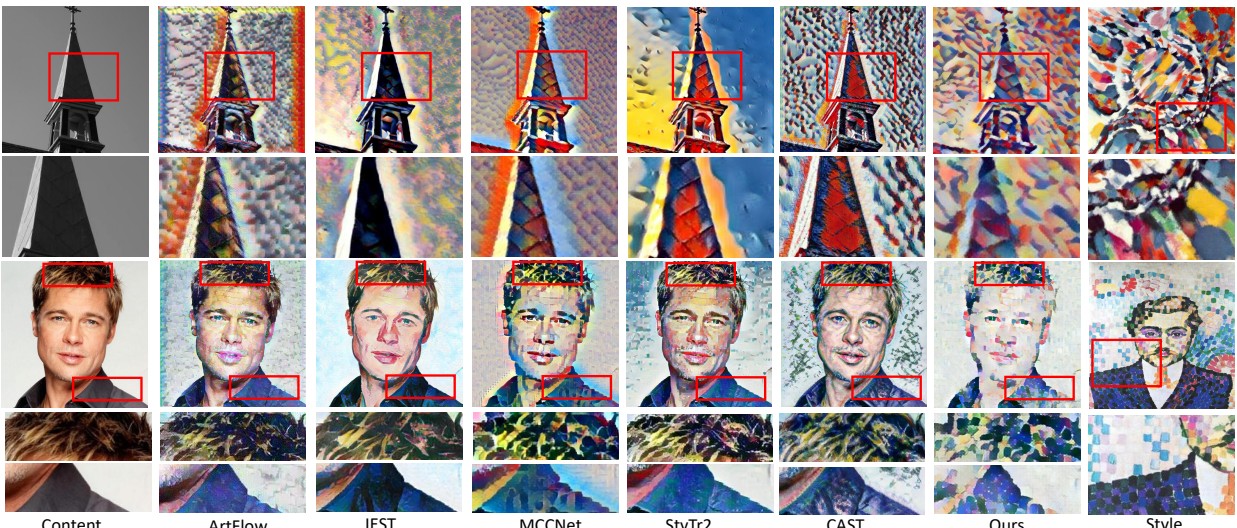

| Content | ArtFlow | IEST | MCCNet | StyTr2 | CAST | Ours | Style |

Figure 7: Comparisons with more state-of-the-art methods in terms of representing details. As discussed in Fig. 6, again our method outperforms the prior at preserving holistic color distributions and variations ($1^{st}$ row), as well as at synthesizing brushstrokes ($1^{st}$ row) and block-wise textures ($2^{nd}$ row).

row) and often repeats undesirable patterns in smooth regions (e.g. distorted artifacts in the background of the $3^{rd}$ and $5^{th}$ row). Moreover, SANet tends to blur out edges (e.g. the $4^{th}$ and $5^{th}$ row). Looking closely at the details in the last row, we see both SANet and AdaAttn copies the eye over the entire content image (highlighted by the red arrows).

In parallel, in Fig. 7, ArtFlow (An et al., 2021) and CAST (Zhang et al., 2022) consistently produce unnatural artifacts over smooth background while IEST (Chen et al., 2021) and StyTr2 (Deng et al., 2022) are weak in discerning significant textures (e.g. the colorful brushstrokes on $1^{st}$ row and blob-wise patterns on $2^{nd}$ row). All methods including MCCNet (Deng et al., 2021) present unseen halations.

Our method generalizes well to a multitude of styles, from holistic color distribution (e.g. skies with diverse colors in the $4^{th}$ row) to local brush strokes (e.g. the last row) and detailed textures (e.g. block-wise patterns in the $2^{nd}$ and $5^{th}$ row). Therefore we have demonstrated that our method can catch more style ingredients while faithfully preserving details from target styles. Meanwhile, the CCLoss allows us to significantly reduce various artifacts (e.g. the $3^{rd}$ row) and render sharp contours in all cases. We can reach the same conclusion in Fig. 7 for ArtFlow, IEST, MCCNet, StyTr2 and CAST.

**Quantitative comparison.** Following Li et al. (2017); Song et al. (2019); Deng et al. (2022), we assess our method against baselines quantitatively with the perceptual style loss. We randomly pick 20 content images and 30 style images from our test set to synthesize 600 stylized images, and list losses in the first row of Tab. 1. We see 1) CCNet can already achieve a decent style loss even if optimized with the CCLoss only; 2) when trained with both CCLoss and style loss, CCNet yields a remarkably lower style loss than other methods: $L_s = 0.4634$, $L_{com} = 0.3087$ and $L_{coh} = 0.3058$. These results confirm that explicitly modelling completeness and coherence allows CCNet to capture style patterns more efficiently than the baselines.

Also, following (Chen et al., 2021; Zhang et al., 2022) we use the Deception score to measure the closeness between synthesized images and images created by human artists. Specifically, we take the 30 style images as reference, then for each method compute the percentage of stylized outcomes mistaken as human-created. We collect 3,000 responses from 30 participants and use the average votes as the measurement. We also compute the CCLosses defined in Eq. 11 and Eq. 12 to quantitatively assess coherence and completeness. We see CCNet achieves the lowest scores on the three metrics, comprehensively demonstrating its superior performance over its predecessors.

In the $4^{th}$ row of Tab. 1, we report the average running time of each method over 100 test images. Gatys et al. (2016) is the slowest due to its slow optimization regime. SANet achieves the fastest speed due to

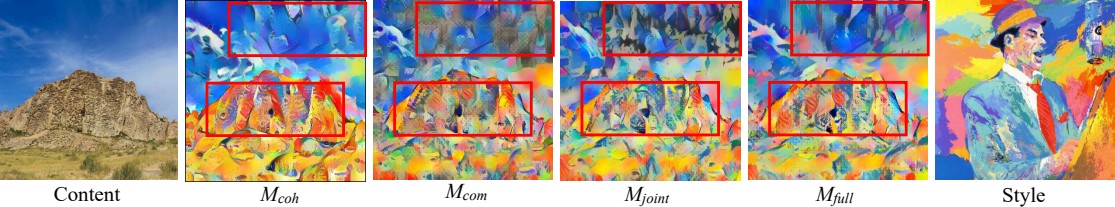

Figure 8: Ablation study on completeness and coherence modeling as well as joint analysis. $M_{full}$ denotes the unablated CCNet. $M_{joint}$ refers to the CCNet with Joint Analysis Attention Modules removed; $M_{coh}$ refers to the CCNet with the Completeness Attention Module removed; $M_{com}$ refers to the CCNet with the Coherence Attention Module removed.

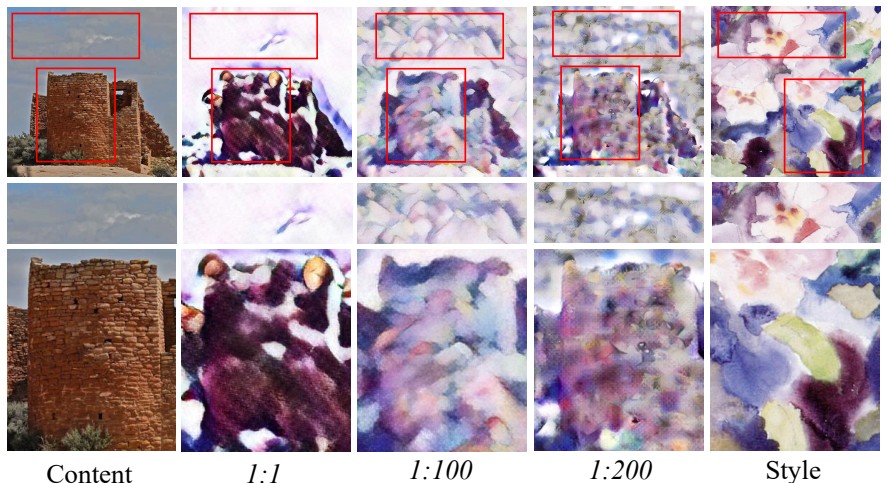

Figure 9: Results obtained by gradually biasing stylization towards coherence. A larger coherence-over-completeness ratio allows the model to capture style patterns (e.g. the dominated textures in the sky and wall) more completely, but would lead to more distorted patterns (e.g. the undesirable gray in the sky and blurry textures on the wall).

its simplicity; WCT and AvatarNet take more running time since they both require the SVD operation. CCNet's running time is fairly competitive when compared with the baselines, and in fact comparable to that of SANet (Park & Lee, 2019).

We then conduct a user study to assess the visual appeal of images rendered by all methods. First, we randomly select 25 content images and 30 style images from the test dataset, and create their complete combination to synthesize 750 stylized images for each method. For each participant, 10 stylized images of CCNet and one of the other methods are displayed side-by-side in a random order. Thus, each participant needs to make 90 votes for 9 baseline methods and we receive 3150 response in total. During user study, the participants were asked to choose the image that learns the most characteristics from the style image. Specifically, the participants were told that the preservation of significant style patterns was the primary evaluation point. Additionally, the assessment time is longer than 30 seconds for each question so that each participant can make careful decision. We list their preference scores in the last row of Tab. 1. The Fig. 20 further illustrates the age and gender distributions of the 35 participants. It can be seen that, the ages and genders of the participants spread uniformly, which clearly indicates the validity and unbiasedness of our experimental settings.

**Ablation study.** To investigate the impact of completeness and coherence modelling to the performance of our method, we first remove the Joint Analysis Attention Modules from our pipeline, so the network would not seek for the best compatibility between completeness and coherence. From Fig. 8 we see the ablated model ($M_{joint}$) distorts textual details and produces a weird black pattern in the sky. Further, we remove the Completeness (Coherence) Attention Module to prevent our pipeline from explicitly imposing the completeness (or coherence) property. We see the model with only the Coherence Attention Module ($M_{coh}$)

Table 1: Following prior work, we assess the methods quantitatively with the perceptual style loss, stylization speed, deception score and user preference. We also use CCLoss to evaluate the completeness ($L_{com}$) and coherence ($L_{coh}$) of rendered images. We measure stylization speed by running time and report them in seconds. The preferences represent the percentage of votes that deems a baseline's result is inferior to ours.

| Loss | Gatys | AdaAttn | WCT | Avatar | SANet | ArtFlow | IEST | MCC | StyTr2 | Ours |
|---|---|---|---|---|---|---|---|---|---|---|
| Style($L_s$) $\downarrow$ | 0.5751 | 1.1561 | 0.5620 | 1.1019 | 0.6215 | 1.0967 | 2.0925 | 1.0710 | 0.7334 | 1.0911 |
| $L_{com} \downarrow$ | 0.3263 | 0.3393 | 0.3293 | 0.3194 | 0.3204 | 0.3478 | 0.3530 | 0.3356 | 0.3366 | 0.2979 |
| $L_{coh} \downarrow$ | 0.3332 | 0.3255 | 0.3397 | 0.3150 | 0.3222 | 0.3501 | 0.3376 | 0.3315 | 0.3306 | 0.2795 |
| Speed $\downarrow$ | 56.81 | 0.1083 | 0.4952 | 0.6772 | 0.0716 | 0.6139 | 0.0914 | 0.0758 | 0.7453 | 0.0908 |
| Deception score | 0.36 | 0.30 | 0.39 | 0.40 | 0.44 | 0.27 | 0.29 | 0.24 | 0.30 | 0.54 |
| Preference | 0.814 | 0.740 | 0.780 | 0.749 | 0.657 | 0.786 | 0.723 | 0.791 | 0.700 | - |

cannot render the diverse set of styles from the original style images (e.g. colors in mountainous areas). While the model with the Completeness Attention Module only ($M_{com}$) retains more salient patterns, it introduces spurious artifacts in the sky. Hence we can confirm completeness and coherence modelling as well as the joint analysis are all essential to the feed-forward process.

We also demonstrate the crucial role of completeness and coherence modelling in training in Fig. 2. CCNet on its own cannot produce visual satisfactory results— see the result produced by CCNet trained on perceptual loss. But our objective function (CCLoss) improves the result notably— patterns become less distorted and edges got sharpened up. We then evaluate the effect of each term in CCLoss by gradually increasing the ratio of coherence weight over completeness weight ($\lambda_{cc}^{coh} : \lambda_{cc}^{com}$). Fig. 9 shows that raising the ratio leads to full style modeling but introduces more distorted patterns. So we conclude that CCLoss allows us to easily strike the balance between completeness and coherence by explicitly tuning the weights.

**Runtime applications.** We demonstrate the flexibility of CCNet by showcasing two runtime applications with a trained stylization model: 1) Adjusting the degree of stylization at runtime (Fig. 10). We can strike a balance by interpolating between two feature maps, $F_{csc}$ and $F_{ccc}$ as: $F_{csc} \leftarrow \alpha F_{csc} + (1-\alpha)F_{ccc}, \forall \alpha \in [0,1]$. $F_{ccc}$ represents the output feature given two identical content images as inputs. The network either replicates the content image if we set $\alpha = 0$, or produces a fully stylized image $I_{cs}$ if we set $\alpha = 1$. We show that changing $\alpha$ from 0.2 to 1 leads to a smooth transition. 2) Segmented stylization of an image (Fig. 11). Inspired by WCT (Li et al., 2017), we apply $k$ masks $M = \{M_1, M_2, \cdots, M_k\}$ to indicate the spatial correspondence between $k$ image regions and desired styles $\{I_1, I_2, \cdots, I_k\}$. Then we extract a specific content region, as $\hat{I}_c^i = M_i \odot I_c$, where $\odot$ denotes a simple mask-out operation. Next, we achieve the specified stylized image $\hat{I}_{cs}^i$ by feeding $\hat{I}_c^i$ and $I_s^i$ into CCNet as $\hat{I}_{cs}^i = CCNet(\hat{I}_c^i, I_s^i) \odot M_i$. The final spatially-controlled stylization $I_{cs}$ is produced by combining all $\{\hat{I}_{cs}^1, \hat{I}_{cs}^2, \cdots, \hat{I}_{cs}^k\}$ as $I_{cs} = \sum_i \hat{I}_{cs}^i$.

# 5 Conclusion, Limitations, and Societal Impact

We make the first attempt to explicitly define completeness and coherence for style transfer in a patch-based manner. We propose the Completeness and Coherence Network (CCNet) as well as the CCLoss to explicitly impose the two objectives and model their inter-dependency. CCLoss allows us to strike a balance between the two objectives and to generate vivid stylization details. Experimental results show that compared with prior work, our method renders a richer set of styles while preserving fine-grained details.

Content ⟶ Style

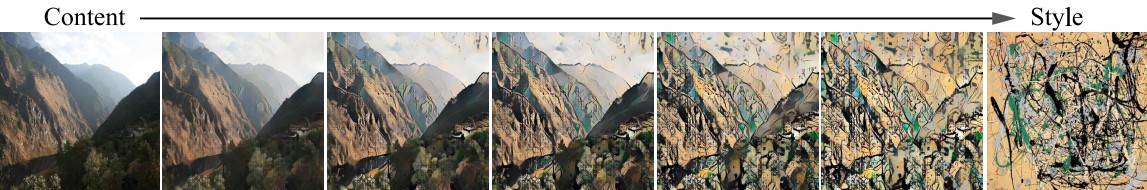

Figure 10: We can explicitly balance between content and style by tuning $\alpha$.

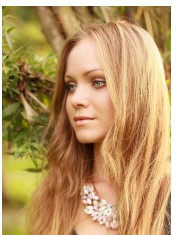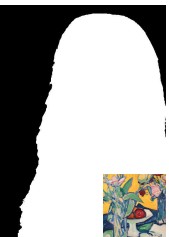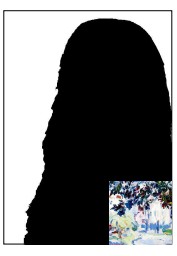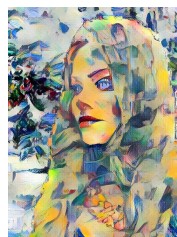

Figure 11: An example of spatially-controlled stylization via mask-out operations.

**Limitations.** One limitation of this work is that our CCNet is patch-based, thus is relatively weaker than statistics-based alternatives in capturing and augmenting information at different scales. Another limitation is that the complexity of the computed affinity matrices is $O(n^2)$, making CCNet limited to a pair of $2048 \times 1024$ images on a TitanX GPU with 12GB memory. But these limitations can be addressed in future work. One can potentially draw inspiration from AvatarNet (Sheng et al., 2018) or simply merge features with different patch sizes for multi-scale communication. Other interesting directions include approximating the affinity matrices (Zhu et al., 2019) and extending our main idea to other related fields, like image translations and texture synthesis.

**Societal impact.** Our work can lead to more efficient image editing pipelines. While such algorithms can empower artists to author more creative contents, they could be used maliciously, e.g. for creating fake portraits on social media. Nonetheless, recent work such as Wang et al. (2020) has demonstrated great potential in detecting fake images produced by powerful generators (Karras et al., 2019; 2020), therefore can ameliorate the concern to some extent.

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

## A   Appendix: Overview

In Sec. B, first of all we showcase more experimental results to demonstrate that modelling completeness and coherence allows our method to effectively render fine details. Then we present more results from our ablation study on the impact of joint analysis, completeness and coherence modeling, patch size as well as multi-scale embedding. Next, in Sec. C we detail the implementation of our Non-local Diffusive Attention Module (Wang et al., 2018), and we present a lightweight version of this module in Sec. D. Next, in Sec. E we present additional qualitative results on SANet (Park & Lee, 2019) to compare with CCNet in terms of synthesis quality and design paradigms. From Sec. F to Sec. H, we use a stylization matrix and samples of stylized high-resolution images to demonstrate that our method generalizes well. We also include a discussion on the applicability of various contents in Sec. H. Note that the MS-COCO (Lin et al., 2014) and WikiArt (Nichol, 2016) dataset in our experiments are public and under a Creative Commons Attribution 4.0 License, which permits us to distribute, remix, tweak, and build upon them.

## B   More Experimental Results

In this section, we present more evaluation results about completeness and coherence tradeoffs. We also show more results from our ablation study experiments to verify the joint analysis paradigm, the completeness and coherence modeling, as well as multi-scale embedding.

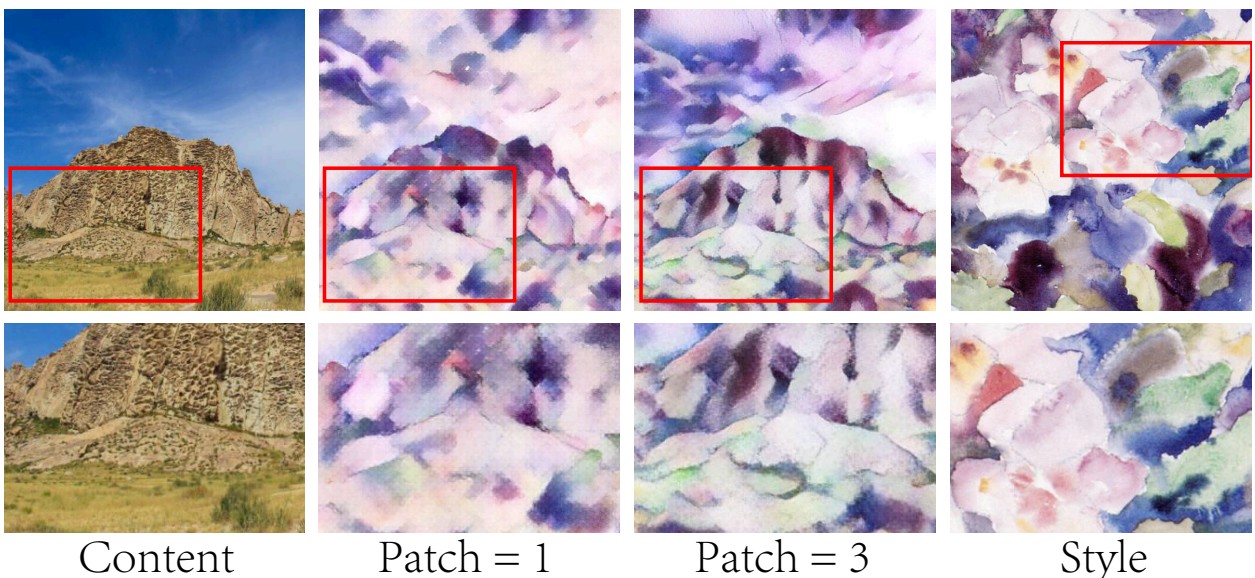

Figure 12: Ablation study on the impact of patch size.

**Comparison with previous methods on synthesizing details.** In order to demonstrate our capability in generating detailed style patterns, we attach close-up views of figures from the main text: in Fig. 13, we show that the CCNet outperforms prior methods on reducing repetitive patterns, lowering distorted clutters (e.g. $1^{st}$ row) and rendering clear contours (e.g. $3^{rd}$ row). The CCNet also excels at synthesizing detailed textures (e.g. the patterns in $2^{rd}$ row) and improving style diversity (e.g. the brush strokes in the sky of $1^{st}$ and $2^{rd}$ rows).

**Trading off between completeness and coherence.** Our CCLoss plays a pivotal role in improving the synthesis of details and trading off between completeness and coherence. We can explicitly balance between completeness and coherence by adjusting the ratio between the coherence and completeness term ($\lambda_{cc}^{coh} : \lambda_{cc}^{com}$) in CCLoss: from left to right in Fig. 14 ((a) - (f)) are the stylized images generated by six ablated models obtained by gradually increasing the ratio. As the ratio becomes larger, the rendered output manifests more complex style patterns (e.g. the rich color distribution in the mountainous region). But a larger ratio also introduces incoherence: see the light gray clutter in the sky of the $1^{st}$ row.

**The effects of patch size.** Fig. 12 illustrates the influences of patch size on stylization. As mentioned in Sec. 3.1, a larger patch size allows us to more effectively capture coarse-scale structures and model local information, so that we can preserve clearer contours: see the edges within the box containing the mountain and the grassland. Moreover, a model with larger patches can produce more adaptive style details, such as the fine-grained textures in the enlarged region. However, increasing patch size leads to significantly higher computational cost. We set patch size to 1 for our experiments so that we can achieve satisfactory rendering performance while incurring only moderate computational cost.

**Quantitative results from our ablation study.** Aside from the qualitative results in Sec. 4, we show quantitative results on our ablated models in Tab. 2. We see that the computed ablation scores back our claims from the main text. These results further validate the effectiveness of completeness and coherence learning, our proposed joint analysis paradigm and CCLoss.

**Joint analysis.** We use a lightweight version of the Non-local Diffusive Attention Module (Fig. 16) to show that our joint analysis paradigm with non-local blocks is indeed effective. Similar to the canonical structure in Fig. 16 (a), we feed $\tilde{F}_{coh}$ and $\tilde{F}_{com}$ into two $1 \times 1$ convolutional layers respectively. Then we fuse the completeness and the coherence feature via element-wise multiplication. The fused feature is then further

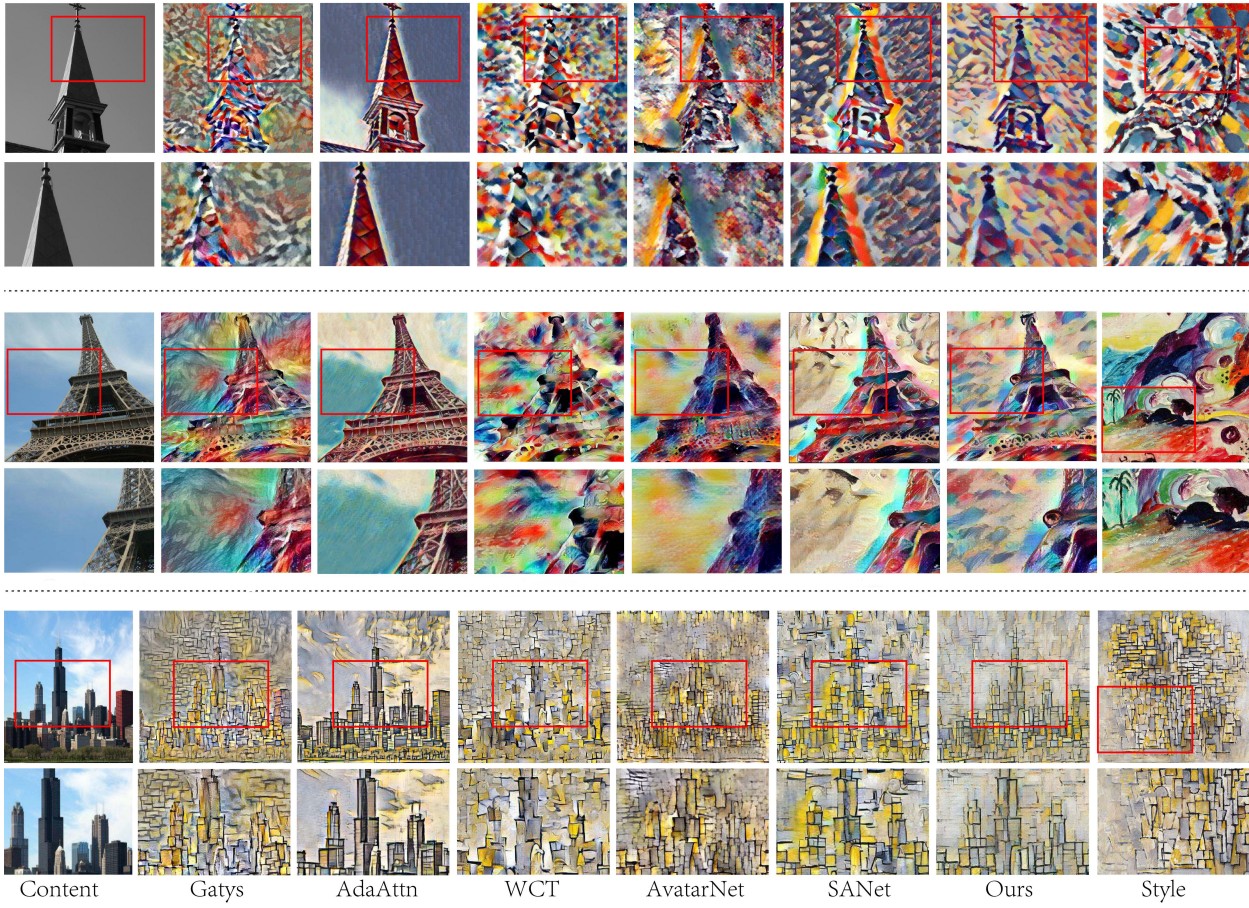

Figure 13: More comparison with existing methods on synthesizing details.

Table 2: Following Tab. 1, we include more quantitative results as a part of our ablation study. We randomly select ten content images from MS-COCO (Lin et al., 2014) and ten style images from WikiArt (Nichol, 2016) to produce 100 stylized images and averaged their metrics. Here $\hat{M}$ refers to the SANet framework (Fig. 2) which contains only the Coherence Attention Module of the CCNet. We use $\hat{M}_{CCLoss}$ and $\hat{M}_{ploss}$ to refer respectively to the model trained with CCLoss and the perceptual loss. We use each of $\{M_{1:1}, M_{1:100}, M_{1:200}\}$ to represent the CCNet trained by setting completeness and coherence ratio ($\lambda_{cc}^{coh}:\lambda_{cc}^{com}$) to 1 : 1, 1 : 100 and 1 : 200 respectively. We carry the notation $\{M_{coh}, M_{com}, M_{joint}\}$ over here from Fig. 8 to demonstrate quantitatively the effectiveness of each attention module in CCNet. Note that $\hat{M}_{CCLoss}$ equals to $M_{coh}$ here.

| Loss | $M_{coh}$ | $M_{com}$ | $M_{joint}$ | $M_{1:1}$ | $M_{1:100}$ | $M_{1:200}$ | $\hat{M}_{ploss}$ | $M_{ploss}$ | $\hat{M}_{CCLoss}$ | $M_{full}$ |
|------|-----------|-----------|-------------|-----------|-------------|-------------|-------------------|-------------|--------------------|------------|
| Style($L_s$) | 192.52 | 164.29 | 179.01 | 604.32 | 170.12 | 168.99 | 180.36 | 162.78 | 192.52 | 169.58 |
| $L_{com}$ | 29.36 | 28.74 | 28.76 | 30.67 | 27.82 | 27.65 | 29.03 | 27.59 | 29.36 | 27.93 |
| $L_{coh}$ | 27.01 | 27.99 | 27.87 | 26.86 | 27.25 | 27.97 | 27.74 | 28.21 | 27.01 | 27.19 |

processed by a sigmoid operator, becoming an element-wise weight that weighs how much information of $\tilde{F}_{com}$ should be used to compute the residual feature $\hat{F}_{com}$. Finally, $\hat{F}_{com}$ is also applied to update $\tilde{F}_{com}$ as in the default joint analysis. The element-wise multiplication allows the model to select compatible features between completeness and coherence, so we can employ better-matched data to coherence to update the information from completeness, making $\tilde{F}_{com}$ to be more consistent with $\tilde{F}_{coh}$. By feeding the $\tilde{F}_{coh}$ and $\tilde{F}_{com}$ into the shared non-local block in the opposite direction, we also can enforce $\tilde{F}_{coh}$ to be more similar to $\tilde{F}_{com}$ as shown in Eq. 4 of the main text.

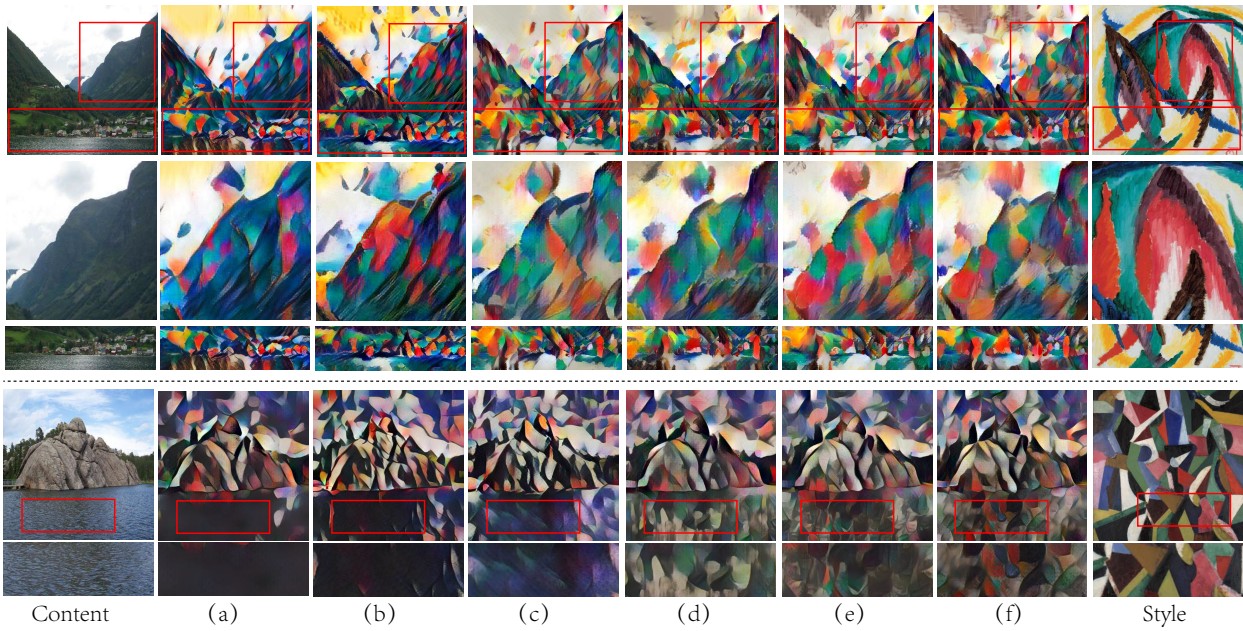

Figure 14: Ablation study for the completeness and coherence term ($\lambda_{cc}^{coh}$ and $\lambda_{cc}^{com}$).

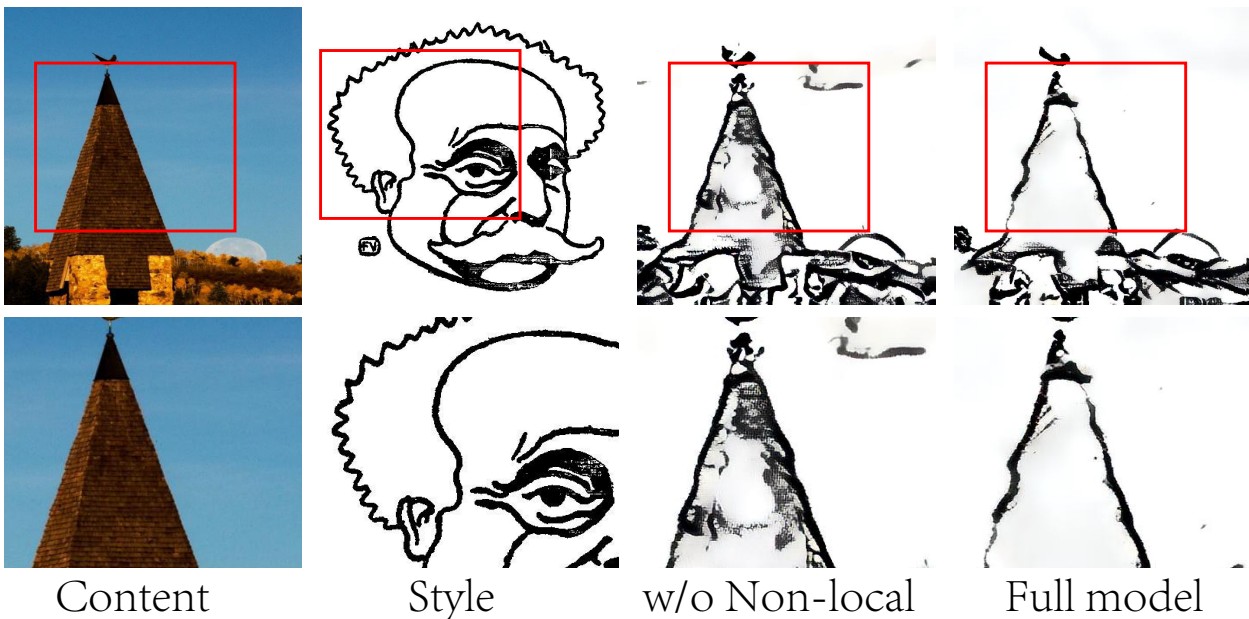

Figure 15: Ablation study for joint analysis.

In Fig. 15, we see the ablated model fails to preserve the texture consistency within a smooth region (e.g. black artifacts within the tower region) due to the lack of long-range dependency modeling. In contrast, the full model is better at presenting the large-scale patterns and reducing repeated artifacts in the background. Note that in addition to element-wise multiplication, we also attempted other fusion operations, such as element-wise addition or computing dot product, but achieved only subtle visual differences on the ablated models.

**Modelling completeness and coherence.** In Fig. 18, we showcase more results from our ablation study on the effectiveness of completeness and coherence as well as on the joint analysis between them. The model with only *Coherence Attention* preserves the content structures well and yields faithful style details, but without completeness modelling it fails to distinguish the overall style distributions. The model with only *Completeness Attention* can introduce more complex style variations to the results, but often generates unseen light black colors in the background regions (e.g. the sky). The model without joint analysis distorts the style patterns, because the model fails to filter out completeness and coherence features that are incompatible. Compared with the ablated models, our full model is better at parsing the complete style components with vivid details.

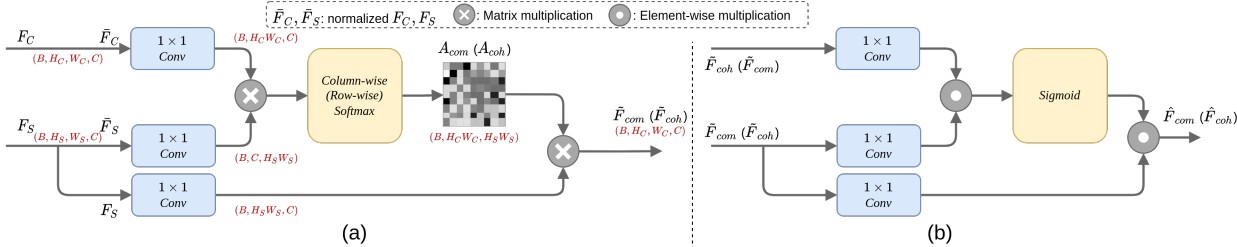

Figure 16: (a) The architecture of our Non-local Diffusive Attention Module. (b) The architecture of an ablated version of (a) which jointly analyzes completeness and coherence but only locally. By replacing the matrix multiplication and softmax operation with an element-wise multiplication (denoted as $\odot$) and a sigmoid operation respectively, we ensure the model only considers the completeness and coherence information on a pair of corresponding pixels, thus impossible to capture long-range dependencies within an entire image. Please refer to paragraph *Joint analysis* in Sec. B and Sec. C for details.

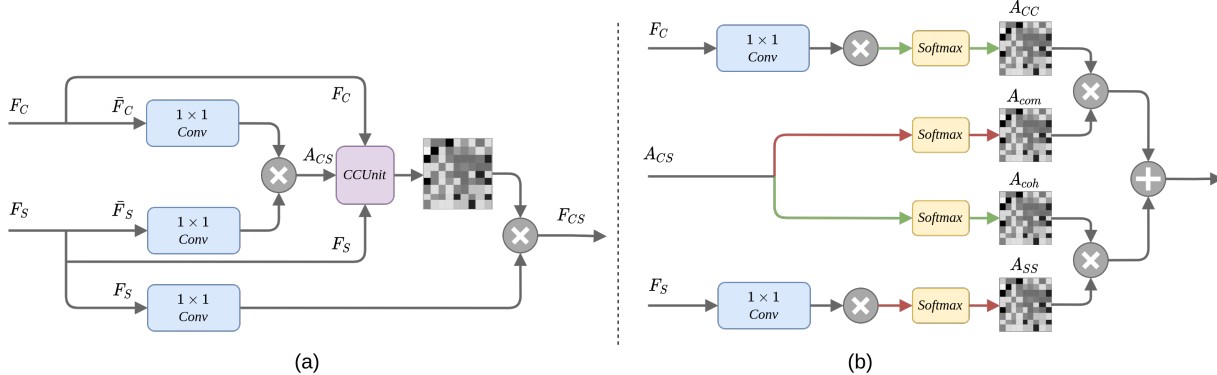

Figure 17: (a) Architecture of the lightweight Non-local Diffusive Attention Module. (b) Architecture of our CCUnit which captures completeness and coherence at once. Please refer to Sec. C for details.

**Multi-scale embedding.** Prior work has used mutli-scale modelling extensively to improve stylization performance by enriching local and global patterns (Sheng et al., 2018; Park & Lee, 2019). Here we would like to determine if our multi-scale approach is effective. So we remove the branch for $relu\_4$ and $relu\_5$ respectively. As shown in Fig. 19, while $relu\_4$ can produce features that entail local color distribution and preserve spatial layouts, it fails to synthesize the circular pattern. But $relu\_5$ is able to render the circular patterns because of its larger receptive field. But $relu\_5$'s content structures are severely distorted and details in the patterns are blurred out. By integrating these two scales into our model, we can capture richer salient style patterns and maintain the content structures simultaneously, yielding better stylization results.

**Stability for quantitative metrics.** In order to measure the data stability of computed metrics, we report the error bars in Fig. 21 for the $L_s$, $L_{com}$, $L_{coh}$ and preference scores used in Tab. 1. For $L_s$, $L_{com}$

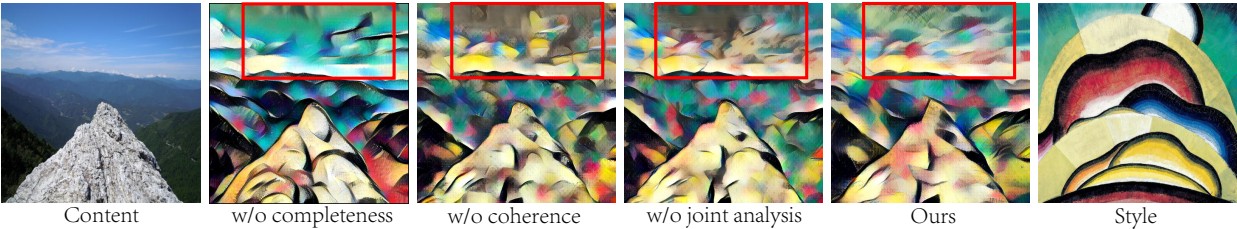

Content w/o completeness w/o coherence w/o joint analysis Ours Style

Figure 18: Ablation study on modelling completeness and coherence.

and $L_{coh}$, the mean and standard deviation in each bar are computed based on the images produced by a specific method. For example, we compute the mean and standard deviation of the 600 stylized images' $L_{com}$ for Gatys et al. and report the result in the first column of Fig. 21 (a). We also list the 90% confidence intervals in Tab. 3-4 to directly indicate the representative level of results. As for the preference score, however, each baseline's standard deviation is computed over the votes from each participant (Fig. 21 (d)) and votes corresponding to each method (Fig. 21 (e)). It is seen that, the produced scores are quite stable, showing the strong representation capability of the selected image set.

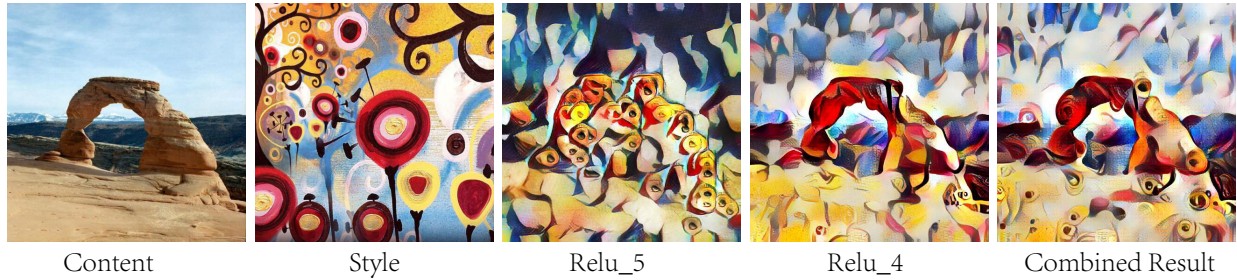

Content Style Relu_5 Relu_4 Combined Result

Figure 19: Ablation study on multi-scale embedding.

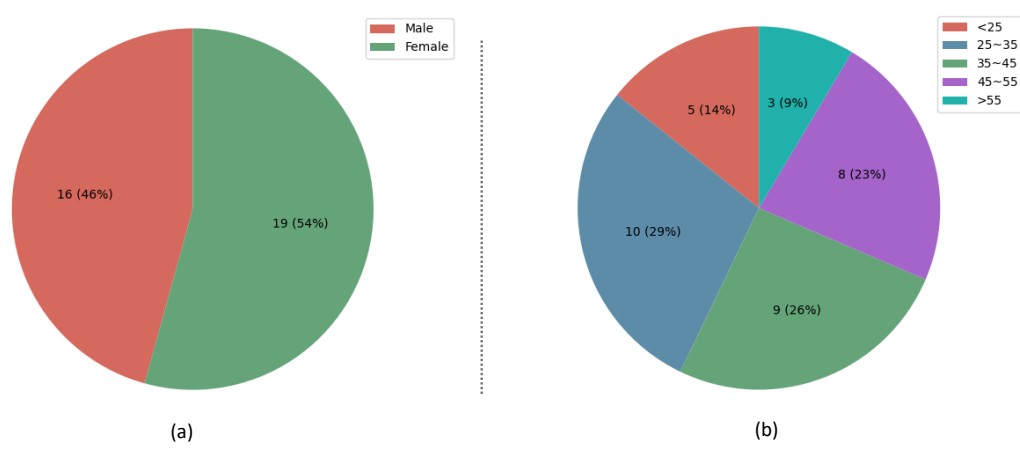

Figure 20: In the user study, the gender (a) and age (b) of the participants distribute uniformly, showing the unbiasedness of the conducted evaluation. See the user study part in Sec. 4 for more detailed discussion.

Lastly, in Fig. 22 we show stylization results with more inputs on the following baselines: ArtFlow (An et al., 2021), IEST (Chen et al., 2021), MCCNet (Deng et al., 2021), StyTR2 (Deng et al., 2022), CAST (Zhang et al., 2022). Still our method outperforms the rest at capturing the complete style without leaving spurious artifacts, further proving our empirical advantages.

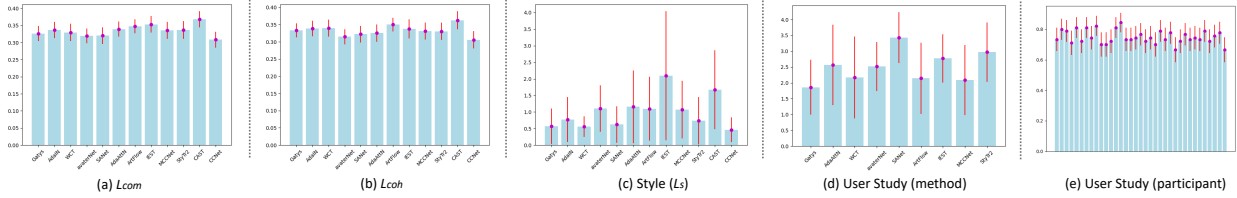

| (a) $L_{com}$ | (b) $L_{coh}$ | (c) Style ($L_s$) | (d) User Study (method) | (e) User Study (participant) |

Figure 21: The error bars for $L_{com}$, $L_{coh}$, $L_s$ and preference scores used in Tab. 1. Note that (d) presents the average votes computed for each method while (e) presents each participant's vote percentages which prefer our results.

Table 3: Confidence intervals for $L_s$, $L_{com}$ and $L_{coh}$ to show the quantitative result stability (part 1).

| Loss | Gatys | AdaAttn | WCT | Avatar | SANet | Ours |
|---|---|---|---|---|---|---|
| Style($L_s$) ↓ | $0.5751 \pm 0.0365$ | $1.1561 \pm 0.0748$ | $0.5620 \pm 0.0214$ | $1.1019 \pm 0.0479$ | $0.6215 \pm 0.0377$ | $1.0911 \pm 0.0258$ |
| $L_{com}$ ↓ | $0.3263 \pm 0.0015$ | $0.3393 \pm 0.0015$ | $0.3293 \pm 0.0017$ | $0.3194 \pm 0.0005$ | $0.3204 \pm 0.0016$ | $0.2979 \pm 0.0016$ |
| $L_{coh}$ ↓ | $0.3332 \pm 0.0014$ | $0.3255 \pm 0.0017$ | $0.3397 \pm 0.0017$ | $0.3150 \pm 0.0005$ | $0.3222 \pm 0.0016$ | $0.2795 \pm 0.0017$ |

Table 4: Confidence intervals for $L_s$, $L_{com}$ and $L_{coh}$ to show the quantitative result stability (part 2).

| Loss | ArtFlow | IEST | MCC | StyTr2 | Ours |
|---|---|---|---|---|---|
| Style($L_s$) ↓ | $1.0967 \pm 0.0657$ | $2.0925 \pm 0.1481$ | $1.0710 \pm 0.0597$ | $0.7334 \pm 0.0608$ | $1.0911 \pm 0.0258$ |
| $L_{com}$ ↓ | $0.3478 \pm 0.0014$ | $0.3530 \pm 0.0017$ | $0.3356 \pm 0.0017$ | $0.3366 \pm 0.0018$ | $0.2979 \pm 0.0016$ |
| $L_{coh}$ ↓ | $0.3501 \pm 0.0013$ | $0.3376 \pm 0.0019$ | $0.3315 \pm 0.0017$ | $0.3306 \pm 0.0017$ | $0.2795 \pm 0.0017$ |

## C   More Discussion on the Non-local Diffusive Attention Module

As we have mentioned many times, the primary goal of our method is achieving completeness and coherence. To this end, we introduce a bi-directional patch-based similarity measure to quantify the visual relationships between one stylized result and an input style image. As mentioned in the main text, the similarity optimization procedures inspire us to update the content features by diffusing style information with two different affinity kernels. Specifically, we implement this motivation with the non-local blocks (Wang et al., 2018), as shown in Fig. 16 (a). And we set the patch size to 1 here.

For the diffusion modules in Fig. 16 (a), we take content feature $F_c$ and style feature $F_s$ as inputs whose shapes are $B \times H_c \times W_c \times C$ and $B \times H_s \times W_s \times C$ respectively. Here $B$, $H$, $W$ and $C$ indicate batch size, height, width and channel dimension individually. We first feed the normalized content feature $\bar{F}_c$ and style feature $\bar{F}_s$ into the Attention blocks, followed with two different $1 \times 1$ convolutions. Then we compute the similarities between the normalized $\bar{F}_c$ at one pixel and the normalized $\bar{F}_s$ at another pixel to make a content semantics-based style diffusion (Sheng et al., 2018). And the closest neighbor search for each pixel in $\bar{F}_c$ and $\bar{F}_s$ can be approximated by a softmax performed in different axes. After the *reshape* operation, the shape of the similarity matrix is $B \times H_c W_c \times H_s W_s$. Specifically, let each row of the similarity matrix represent relationships between one pixel in $\bar{F}_c$ and all the pixels in $\bar{F}_s$. Correspondingly, each column indicates the affinities between one pixel in $\bar{F}_s$ and all the pixels in $\bar{F}_c$; refer to Fig. 5 in the main text for details. When we search the closest feature vector in $\bar{F}_s$ for each feature vector of $\bar{F}_c$, we should perform the softmax operation for the similarity matrix along each row. In the opposite direction, we should perform the softmax along each column to search the closest feature vector in $\bar{F}_c$ for each feature vector of $\bar{F}_s$. Finally, the stylized feature is realized by multiplying the similarity matrices with the input style feature $F_s$. Note that the affinity matrix for completeness $A_{com}$ shares the same size with the affinity matrix for coherence $A_{coh}$.

We can jointly analyze $\tilde{F}_{coh}$ and $\tilde{F}_{com}$ to improve their pixel-level compatibility in the same way. Specifically, we compute a residual feature $\hat{F}_{coh}$ as a weighted sum of feature vectors in $\tilde{F}_{coh}$ by simultaneously considering each feature vector of $\tilde{F}_{com}$ and feature vectors at all positions of $\tilde{F}_{coh}$. Then $\hat{F}_{coh}$ is similar to $\tilde{F}_{com}$ at each of their positions. In other words, we rearrange feature vectors of $\tilde{F}_{coh}$ ($\tilde{F}_{com}$) to fit $\tilde{F}_{com}$ ($\tilde{F}_{coh}$) together well; see Eq. 4 - 5 of the main text for details. Then the refined feature $\ddot{F}_{coh} = \tilde{F}_{coh} + \hat{F}_{coh}$ ($\ddot{F}_{com} = \tilde{F}_{com} + \hat{F}_{com}$)

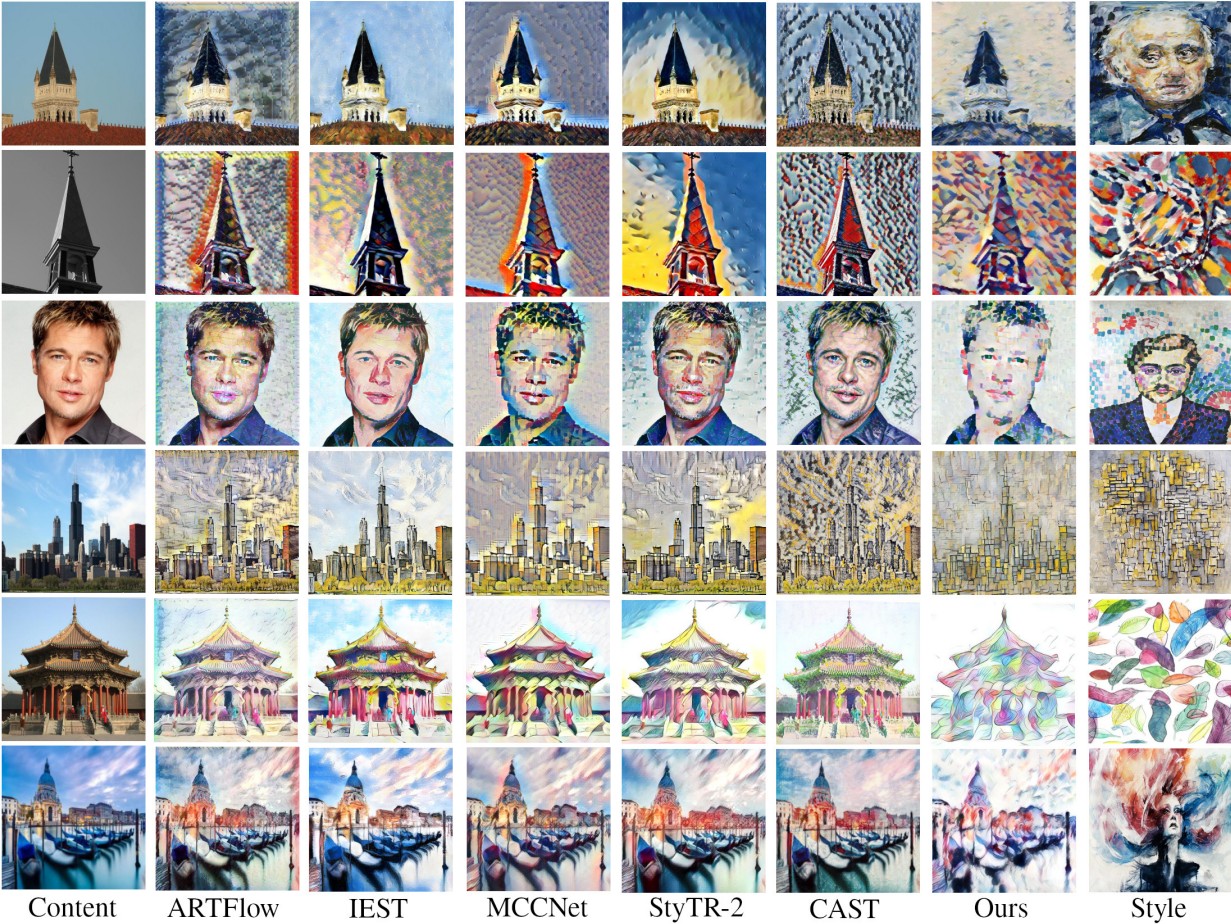

| Content | ARTFlow | IEST | MCCNet | StyTR-2 | CAST | Ours | Style |

Figure 22: Comparisons with the baselines using the same input images as Fig. 6.

will be more similar/compatible to $\tilde{F}_{com}$ ($\tilde{F}_{coh}$) than $\tilde{F}_{coh}$ ($\tilde{F}_{com}$). To verify this joint analysis scheme, we conduct another ablation study experiment; check out the paragraph on *joint analysis* in Sec. B for more information. Hence with the non-local block, it is effective for our network to model the completeness and coherence and capture the long-range dependency between pixels.

# D Simplified Non-local Diffusive Attention Module

In order to further demonstrate the design flexibility of our completeness and coherence concepts, we derive a lightweight version of the Non-local Diffusive Attention Module from its canonical counterpart (Fig 16 (a)). As shown in Fig. 17 (a), we simplify by unifying the two attention layers for completeness and coherence modeling and one attention layer for joint analysis into one simple block. Compared to the heavyweight architecture, the simplified one has its softmax layer replaced with a learnable CCUnit to output the affinity matrix for further feature diffusion. Fig. 17 (b) shows the architecture of the CCUnit which captures completeness and coherence at once. Taking $F_c$, $F_s$ and $A_{cs}$ as inputs, CCUnit feeds $F_c$ and $F_s$ into two $1\times1$ conv layers respectively and compute the corresponding self-similarity matrices (denoted as $\otimes$ in purple). Then CCUnit applies softmax along each axis to compute the relative correlations between one pixel and all the pixels in $F_c$ (or $F_s$) and output $A_{cc}$ and $A_{ss}$. Specifically, the red arrow and green arrow indicate the $1^{th}$ and $2^{th}$ axes correspondingly. Still we apply softmax along one axis to learn completeness and along another to learn coherence. The leared features are $A_{com}$ and $A_{coh}$. We further multiply $A_{cc}$ with $A_{com}$ and $A_{coh}$ with $A_{ss}$ (denoted as $\otimes$ in gray) so that we can explicitly model the long-range dependencies

in input content-style image pairs and better preserve the content structures and coarse style patterns in inputs. Note that the matrix multiplication between $A_{cc}$ and $A_{com}$ ($A_{coh}$ and $A_{ss}$) implicitly corresponds to the functionality of the joint analysis attention layer in main text. It is because joint analysis stage of CCNet aims to implicitly recover the repetitive information in content images for completeness modeling and enrich the style diversity in style images for coherence modeling. Finally we add the two output matrices and obtain the affinity matrix to diffuse style features.

Note that the output affinity matrix of the simplified CCNet can be used to further reduce the computation cost by incorporating it into the multi-scale procedure. Specifically, the output affinity matrix from coarser scale enables the adaptation of style features at finer scales. Despite these extra bonus, they are beyond our focus to verify the effectiveness of completeness and coherence.

Fig. 23 illustrates stylization results produced by the simple version of CCNet to further evaluate the effectiveness of completeness and coherence modeling. All input images can be found the main text. One can see that, our simplified CCNet can maintain clear contours of the prominent objects (e.g. outlines of all the buildings) and introduce faithful style details (e.g. brush strokes in $1^{st}$ row and the block-wise textures in $2^{nd}$ and $3^{rd}$ rows) concurrently. It consistently matches the advantages presented in our full model. In contrast, SANet highly biases towards content structures during aligning style features so that it repeats undesired patterns in smooth regions (e.g. gray distorted artifacts in the background of $1^{st}$ and $3^{rd}$ rows). It often introduces some unwanted halation around edges as well (e.g. $1^{st}$, $2^{nd}$ and $3^{rd}$ rows) and sometimes suffers from incapability to capture dominanted style patterns (e.g. lack of block-wise textures in the temple region of $3^{rd}$ row).

## E  Comparing with SANet on Synthesizing Details

As for style transfer, we make first efforts to associate bi-directional Chamfer matching with non-local blocks (Wang et al., 2018). Compared to other patch-based methods, this perspective introduces some unique advantages to our method. Specifically, the completeness and coherence modeling enables us to (a) explicitly capture these two constraints, (b) minimize artifacts in results, (c) reduce chances to repeat similar patterns in smooth regions, (d) control the effects of completeness and coherence in stylizations, (e) measure the stylization ability via a newly developed metric. Actually, the Style-Attention Network (SANet) (Park & Lee, 2019) can be regarded as an ablated version of our framework, by only keeping the coherence branch but removing the joint analysis module and completeness branch; see more discussions in Sec. 3.2 of the main text. In Fig. 24, we exhibit additional close-up views to further compare the synthesized details of SANet.

As mentioned, SANet does not impose the completeness and is strictly biased to the content features during stylization, which prevents it from considering the interactions between various style elements. Thus it might distort the local textures in smooth regions (e.g. in the sky and surface of the sea) and the detailed variation (e.g. the roof on the top left corner does not preserve the texture arrangements). It also copies repeated patterns to results (e.g. the eyes spread over the whole image). Without explicit constraints on stylization results, the style patterns in background are prone to be distorted (e.g. the black clutters in the sky).

In contrast, our model can better capture the notable colors and dominated textures (e.g. cyan and blue textures on the roof and in the sky). Generally speaking, the explicit completeness and coherence modeling enables us to seamlessly reassemble the style elements and work for different kinds of styles, from global hue to local strokes and detailed texture variations. Additionally, the completeness modeling may also help reduce the chances to repeat similar patterns to different regions; hence eyes in the style image only appear once in our results. The necessary content components are well perceived in the stylized images as well (e.g. the outlines of house and mountains).

## F  Style Interpolation Results

We can convexly combine the style patterns from multiple images $\{I_s^k\}_{k=1}^K$ with weight $\{w_k\}_{k=1}^K$ that $\Sigma_{k=1}^K w_k = 1$ such that the stylized feature are $F_{csc} = \Sigma_{k=1}^K w_k F_{csc}^k$. Note that, the superscript $k$ indi-

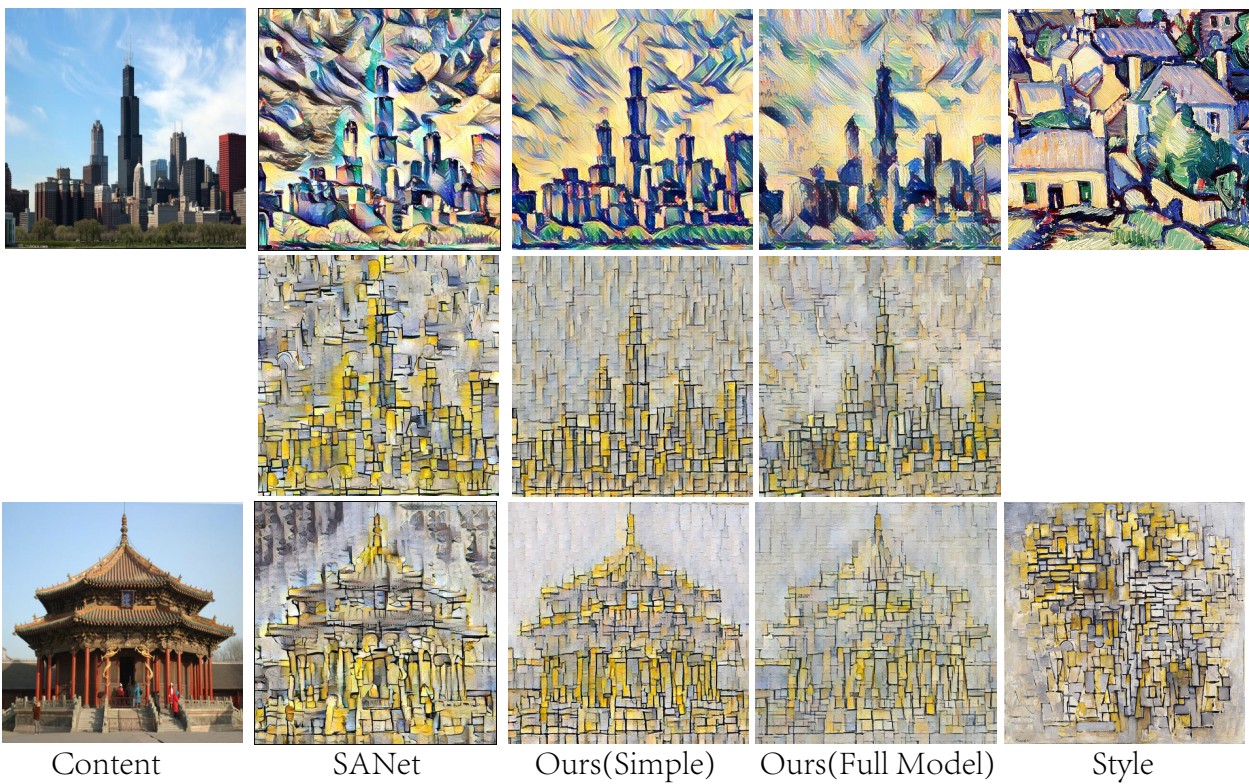

| Content | SANet | Ours(Simple) | Ours(Full Model) | Style |

Figure 23: Comparison between the simplified CCNet, the full CCNet and SANet. We show the input content and style images on the two sides. The simplified CCNet can produce clear contours of prominent objects (e.g. outlines of all the buildings) and yield faithful style details (e.g. brush strokes in $1^{st}$ row and the block-wise textures in $2^{nd}$ and $3^{rd}$ rows), demonstrating the same advantages possessed by our full model. SANet on the other hand biases highly towards content structures while aligning style features, so much so that it repeats spurious patterns in smooth regions (e.g. gray distorted artifacts in the sky of $1^{st}$ and $3^{rd}$ rows). It often introduces halation around edges as well (e.g. $1^{st}$, $2^{nd}$ and $3^{rd}$ rows) and sometimes fail to capture dominant style patterns (e.g. lack of the block-wise textures in the temple region of $3^{rd}$ row).

cates one style image here, but not the feature from $relu\_i$ layer. Finally, the interpolated feature maps are fed into a trained decoder to reconstruct the stylized images, as shown in Fig. 25.

## G  High-resolution Stylization

Here we present an example of high-resolution stylization to demonstrate the applicability to images of large spatial resolutions. As we can see in Fig. 26, the result displays concrete multi-scale style patterns, from the color distributions to faithful style details. Moreover, the texture consistency within different smooth regions (e.g. the sky in the background) is also kept well, clearly displaying the effectiveness of our method for large images. Note that, the usability to large images can be further advanced via the more economical computation strategies for similarity kernels, such as the asymmetric non-local networks (Zhu et al., 2019).

## H  More Discussion

Additionally, one may argue that the patch-based diffusion strategy may constrain the application of our method to input pairs both having similar content structures. But as shown in Fig. 6, the proposed method can generalize to pairs of very different contents by propagating the style information based on the similarities of content structures. It can be realized by performing the normalization before feeding the content feature

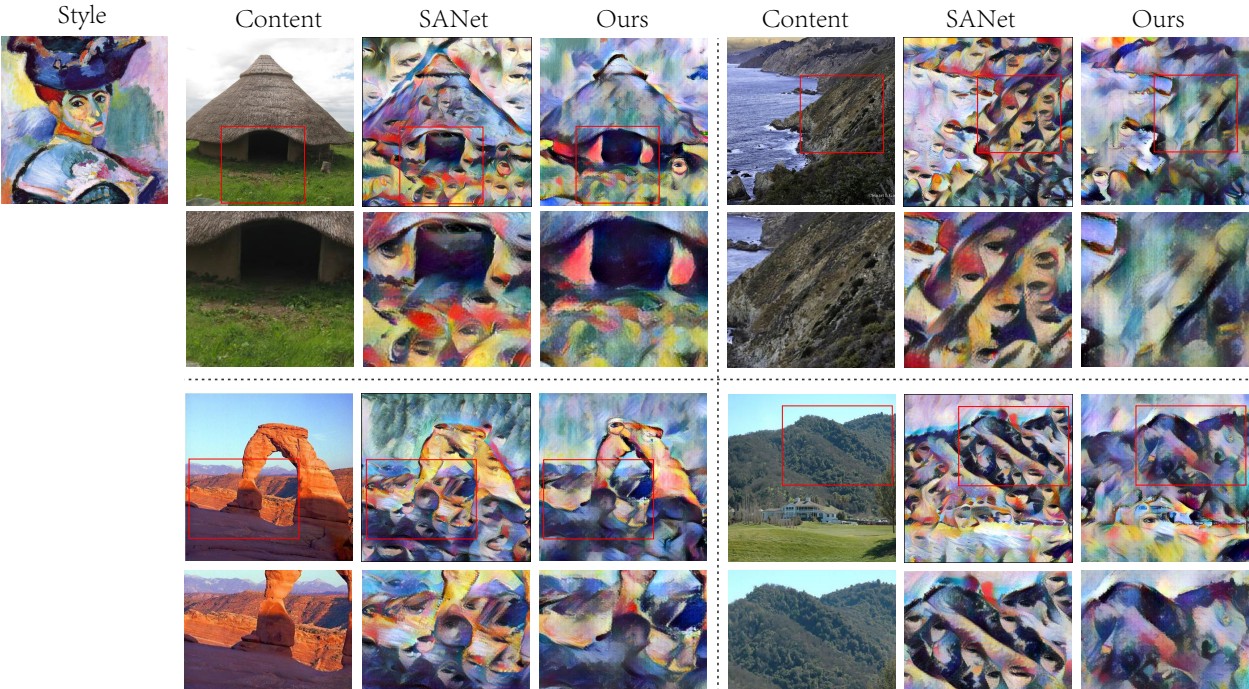

Figure 24: The close-ups for detailed comparisons between SANet and our method.

$F_c$ and style feature $F_s$ into the Completeness and Coherence Attention layers, which shares the same spirit with AvatarNet (Sheng et al., 2018) and SANet (Park & Lee, 2019). During the joint analysis process of the CCNet, the long-range dependency modeling also helps to improve the applicability for different content structures as shown in Fig. 15. Therefore due to the patch-based alignment nature, the proposed method still can present its advantages over the methods based on holistic statistic matching, even when the input image pair differ a lot from each other (Sheng et al., 2018; Park & Lee, 2019).

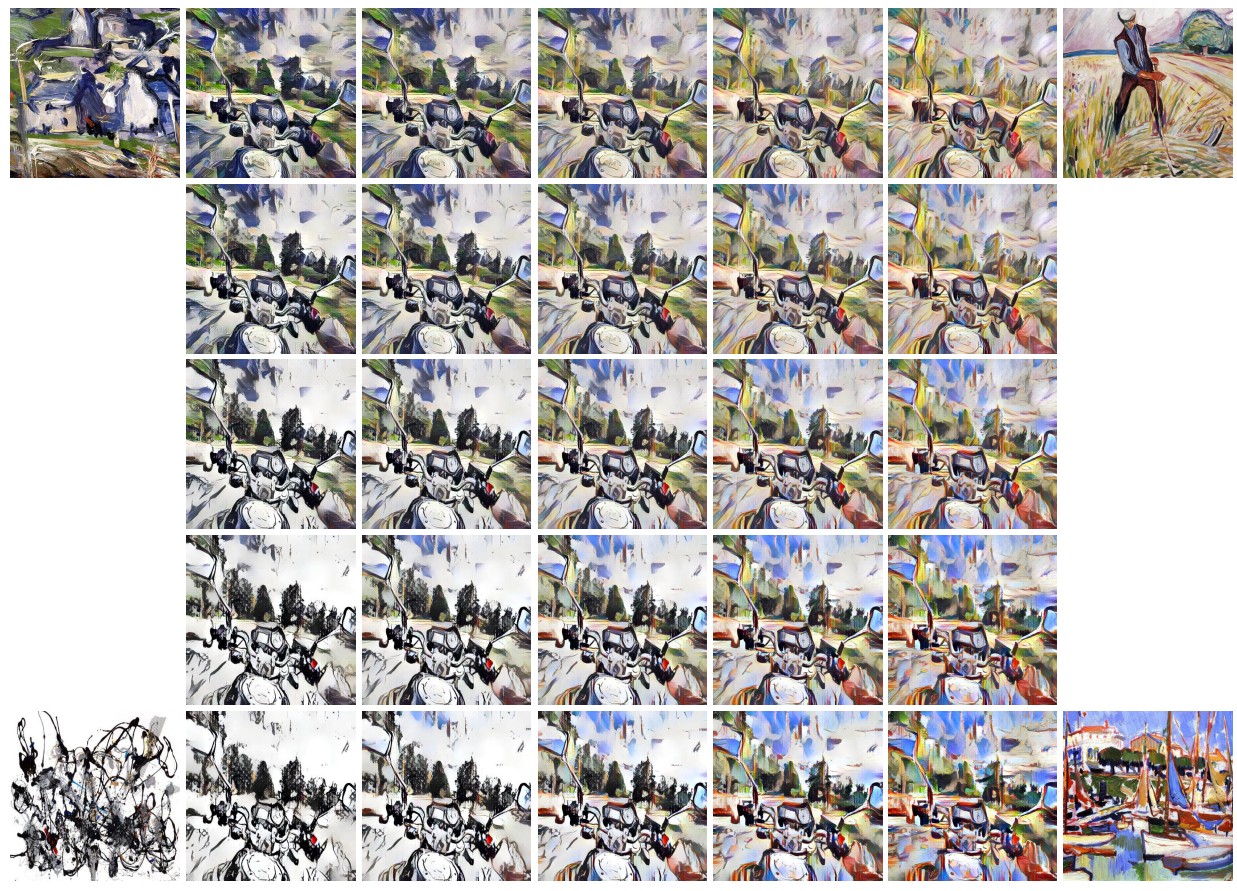

Figure 25: Style interpolation using four different styles.

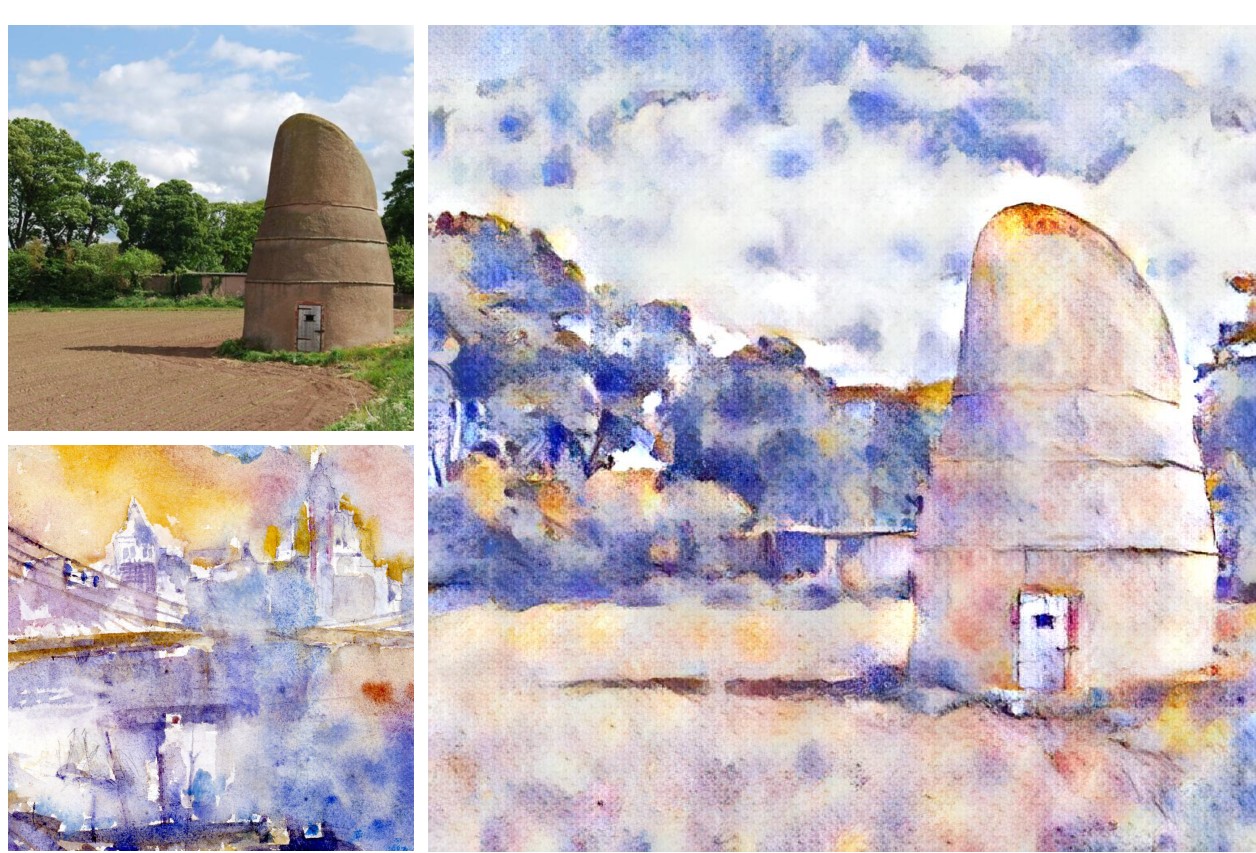

Figure 26: An example of high-resolution stylization. The resolutions of both the content and the style image are $1024 \times 1024$.

