# OpenReview forum: "Completeness and Coherence Learning for Fast Arbitrary Style Transfer"
_TMLR — Accepted by TMLR_

### Review · Reviewer_Hm3y · 2022-07-14

**Summary Of Contributions:**

This work explicitly introduces the concept of completeness and coherence to style transfer. It is claimed that existing methods pursue
the two objectives either partially or implicitly. This work achieves the two objectives explicitly with a novel network design (CCNet) and loss design (CCLoss). The explanation and analysis are easy to follow. Comprehensive experiments have been done to support the claims and the superiority of the proposed technics.

**Broader Impact Concerns:**

The societal impact has been well discussed in this paper.

**Requested Changes:**

There are some minor changes required from my side (discussed in the weaknesses ).

- Please add a reference to Figure 5 when talking about the column&row softmax operation.
- Please explain the style diffusion of $F_S$ and $F_S$ in Section 3.2.
- Please add explanation of symbol $E^i$ in Equation (10).
- Please add more concrete illustrations of style coherence and completeness at the beginning of this paper.

**Strengths And Weaknesses:**

Strengths

- The designs of Completeness and Coherence Network and loss are reasonable for the style transfer task. The usage of column&row softmax operation and attention operation is simple to be applied to other frameworks as well.

- The discussion on SANet is good. It clearly shows the difference between this work and the baseline work and points out the main contributions of this work.

- The limitations including are well discussed. It is claimed that the proposed method is not good at capturing and augmenting information at different scales. There is an ablation study on multi-scale embedding for features relu_4 and relu_5. Why not also involve relu_3?

- Abundant and comprehensive experiments have been done to demonstrate the superiority of the proposed method in terms of completeness and coherence. In Figures 6 & 7, the results and comparisons are well illustrated. The analysis is comprehensive and convincing. In Figure 10, the coherence-completeness loss weight ratio experiment provides clear proof for the claimed effectiveness of coherence loss and completeness loss. More detailed comparisons (including architectures and running time) and ablation studies are also included in the supplementary materials. I would like to see more high-resolution results.

Weaknesses

- The intuition behind the column&row softmax operation is not clear in the beginning. But Figure 5 provides a good explanation. I suggest adding some reference to Figure 5 when talking about the column&row softmax operation.

- In Section 3.2, it is mentioned that “we first normalize $F_S$ and $F_S$ to make a style diffusion based on content structures, yielding $\bar{F}_C$ and $\bar{F}_S$.“ It is confusing to me why the normalization operation can diffuse the style information? The $F_C$ and $F_S$ features seem to be normalized independently, and no statistics are exchanged here.

- The symbol $E^i$ in Equation (10) is not described. Does it mean the VGG encoder? As it is claimed at the beginning of Section 3.3 “VGG-based encoder (denoted as E)”. I suggest describing the symbols more clearly somewhere near the equation.

- The overall organization of this paper is easy to follow. The concepts of style coherence and completeness are not that concrete. I would expect some better illustrations similar to Figure 10 to provide a more concrete example for the readers at the beginning of this paper.

---

> ### Author Response · Authors · 2022-08-02
> **Response to reviewer Hm3y**
>
> Thank you for your helpful comments, and for acknowledging the value of our contributions!
>
> **[The minor presentation changes]**
>
> Thank you for the corrections, we have fixed them in the revision. Specifically, we have added a reference to Fig. 5 at two places of introduction, a description for $E^i$ after Eq. (10) and a new sub-figure in Fig. 1 to exemplify the concepts of completeness and coherence.
>
>
>
> **[The normalization operation in Sec. 3.2]**
>
> Sorry for the confusion about the normalization operation. We clarify that we normalize $F_s$ and $F_c$ not to make a feature diffusion directly. Instead, we aim to remove the texture or other style information in $F_s$ and $F_c$ and denote the output features as $\bar{F}_s$ and $\bar{F}_c$. In this way, the later style feature diffusions are mainly based on the content structures through the similarity computation between $\bar{F}_s$ and $\bar{F}_c$. This point is also discussed in AvatarNet [1] and SANet [2]. We have further clarified this point in the revision.
>
>
>
> **Reference:**
>
> [1]. “Avatar-Net: Multi-scale Zero-shot Style Transfer by Feature Decoration”, CVPR 2018.
>
> [2]. “Arbitrary Style Transfer with Style-Attentional Networks”, CVPR 2019.

---

> > ### Comment · Reviewer_Hm3y · 2022-08-05
> > **Response to authors**
> >
> > Thanks for the authors' effort on the feedback. Most of my concerns have been addressed.
> >
> > The only minor issue from my side is about Fig. 1 (a). It is good to add the illustration of "Full Coherence" and "Full Completeness". I would suggest adding the result of the proposed method to (a) for a direct comparison.

---

> > > ### Author Response · Authors · 2022-08-05
> > > **Thank you for your positive and constructive feedback!**
> > >
> > > Thank you again for your helpful suggestions. We agree that adding the result of proposed method to (a) would be very beneficial to highlight our motivation. We will do it in the revision!

---

### Review · Reviewer_1FEZ · 2022-07-17

**Summary Of Contributions:**

In this paper, the authors explicitly model the concept of completeness and coherence in style transfer and accordingly propose a completeness and coherence network (CCNet) architecture along with a novel CCLoss to balance the completeness and coherence.

**Requested Changes:**

Please see the weakness section for the requested changes.

**Strengths And Weaknesses:**

**Pros:**
- To my knowledge, this is the first work that explicitly models and considers completeness and coherence in stylization.
- The paper is well-written.
- Both quantitative and qualitative results are provided.
- The discussions on limitations and social impact are included.

I’m almost satisfied with this submission, except for some minor concerns:

**Cons:**
- More details on the user study should be provided, e.g., what are the age and gender distributions of the participants?
- It would be great if the authors could consider including more quantitative results such as for example, IS/FID scores.
- The comparison methods are a little bit out-of-date. It would be great if the comparative results of more state-of-the-art methods can be added, like AdaAttN.

---

> ### Author Response · Authors · 2022-08-02
> **Response to reviewer 1FEZ**
>
> Thank you for taking the time to review our paper!
>
> **[More user study details]**
>
> We have supplemented more details about the user study in Sec. 4 of the main text and Fig. 20-21 in the revised appendix. It can be seen that, the age and gender distributions of the participants are nearly uniform, which clearly indicates the unbiasedness and validity of our experimental settings. The instructions of the user study have also been added in the revision.
>
> **[More quantitative results]**
>
> Thank you for this suggestion. As the capability in capturing style patterns is central to stylization methods, we introduce the Deception rate [1] to measure how the synthesized images are akin to the human-created artistic images. Following [1], we use 30 style images as reference and compute the percentage of stylized images that are identified as a real artwork for each method. We report the results of deception rate for the proposed CCNet and other baseline methods in the $5^{th}$ row of Tab. 1.
>
> **[Comparisons with more state-of-the-art methods]**
>
> We respectfully point out that, we have included the latest style transfer methods and reported the qualitative and quantitative results in Fig. 7, Fig. 22 and Tab. 1 respectively. In Fig. 7 and Fig. 22, all the chosen methods are published in 2021 (e.g. ArtFlow, IEST, MCCNet) and 2022 (e.g. StyTr2, CAST). We revise the caption of Fig. 7 to highlight this point.
>
> **Reference:**
>
> [1]. “Artistic Style Transfer with Internal-external Learning and Contrastive Learning”, NeurIPS 2021.

---

### Review · Reviewer_aHVM · 2022-07-22

**Summary Of Contributions:**

The authors propose a method for fast arbitrary style transfer that improves over previous method by providing both completeness and coherence of the resulting stylized image. This means that (a) all style patterns present in the style image are reproduced in the result ("completeness") and (b) the result does not contain any patterns or artifacts not present in the style ("coherence"). The authors claims are

1. Novel loss function that captures both of these objectives
2. Novel network architecture that can produce images that satisfy both objectives
3. Superior performance of their approach to previous methods
4. Optimal balance between running speed, generalization capability and stylization quality


**Broader Impact Concerns:**

No concerns.

**Requested Changes:**

Overall I think this is a great paper. I think the authors could improve it by considering some of the following points.

### Required

1. To enable reproducibility, please describe the 100 images test set used for the quantitative evaluation in Table 1. How were the 100 images selected? Can other authors get access to the same set? How do you ensure that the set is large/representative enough to draw general conclusions about the quality of the methods? What are the error bars / confidence intervals of the metrics on this test set?

1. Similar to above, the user study should be documented better. How were the images selected? Are they a representative, random sample or hand-picked by the authors? How many images (range, average) did each participant rate? What exactly were the instructions to the participants? Please also report error bars on the preference scores

1. The fourth claim of "optimal balance between running speed, generalization capability and stylization quality" (p. 3, top) is not really well defined / supported in my opinion. What constitutes optimality here and why is your method optimal in that sense? If you think about the claim, you're saying that no other method can improve, since yours already achieves the optimal balance. I think this is a minor point that can be fixed by simply dropping this claim.


### Recommended

I leave it to the authors to judge whether they consider the following suggestions useful. I consider none of them mandatory:

1. To what extent is the particular choice of attention modules in CCNet important? While I appreciate the ablation regarding the Joint Analysis Attention, it is possible that the reduced performance simply stems from the fact that removing it makes the network less expressive (shallower). I wonder whether one couldn't simply replace CCNet by a standard Transformer decoder where the encoding of the style image provides the context and the encoding of the content image is processed through multiple standard Transformer decoder layers.

1. Related to above, the ablation results are very qualitative and rely on individual examples. While I generally trust the authors' judgement, I wonder whether you could make this point more quantitative by reporting the same quantities as in Table 1 rows 1–3 for the ablations.

1. The method from Gatys et al. 2016 seems to underperform in this paper. It might be the particular choice of examples images (it's well known that it doesn't deal well with unstructured, uniform background), the selection of styles (a van Gogh or Picasso would be nice), the implementation (no URL given) or just my subjective impression. In my experience, the subjective difference in quality between Gatys and all available fast methods is quite substantial. This difference doesn't come across from the examples shown in the paper, and the user study also seems to suggest Gatys' method is the worst. These two observations make me think that you may not have run Gatys' method with the best settings. Note that if you're using either Leon Gatys' or Justin Johnson's code, you need the old, original VGG checkpoints and cannot use the pretrained ones form torchvision, since the latter have been trained differently and use different preprocessing.

1. Eq (2) uses only a dot product followed by softmax, as is common in Transformer attention modules. However, this formulation is quite sensitive to the contrast of image patches. Since you use the cosine distance for the loss (Eq. 11+12), I wonder whether it would make sense to also use the cosine here?

1. Eq. (7): It seems unnecessarily complicated to include features from both conv4 and conv5. Have you tried using only one of them?

1. Order of Figs. 9 and 10 should be reversed so they appear in the same order as they appear in the text.

1. Fig. 11: The text is missing a lot of important details. For instance, it is not clear where in the pipeline the masking is happening, how that affects the CCnet architecture and attention etc. Please expand the text to provide the necessary details or remove the figure.


**Strengths And Weaknesses:**

### Strengths

+ Impressive results for fast arbitrary style transfer
+ Loss function makes sense and is novel
+ Network architecture (attention module) appears to be suitable
+ Qualitative and quantitative demonstration that results are of high quality
+ Paper is well written and easy to follow



### Weaknesses

- CCNet architecture could potentially be simplified
- User study insufficiently described and reported
- 4th claim not really well defined / supported

---

> ### Author Response · Authors · 2022-08-02
> **Response to reviewer aHVM - part1**
>
> Thank you for your valuable and insightful feedback with a lot of details!
>
> **[Quantitative evaluation discussions]**
>
> First, as mentioned in Sec. 4, we would like to emphasize that how to comprehensively and accurately evaluate the results of style transfer is still an open problem. It becomes even more conspicuous for arbitrary style transfer, as there are no restrictions on content / style image pairs, their scales and other information. So how to perform a quantitative comparison as fair and accurate as possible is an extremely difficult problem. These metrics we are using now are all indirect measures, which have also been agreed by other papers [1, 2]. Therefore, we admit that we cannot completely and accurately draw a general conclusion on the quality of the methods through the existing metrics. Despite this, we make our best efforts to follow the conventions in this field to achieve a general conclusion and highlight our advantages.
>
> Specifically, we follow the previous methods [1, 2] to randomly select 10 content images and 10 style images from the testing dataset to generate 100 stylization images. In this way, we can estimate the generalization capability of each method under different inputs. It follows that the random selection strategy, and the complete combination of loaded content and style images creates large variations between different inputs, which are common in practical situations. Thus, the image selection can best cover our empirical usages.
>
> To further facilitate the generalization capability of our conclusion, we increase the image samples for score computation. Following [1], we randomly apply 20 content images and 30 style images to generate 600 synthesized images. We report the results in Tab. 1 of the revision. It can be seen that, we have much lower $L_{com}$ and $L_{coh}$ than all other methods. However, for the style loss ($L_s$), our CCNet is a bit inferior to some baselines using $L_s$ as the loss function. We assume that it is because our CCNet does not directly train with perceptual loss ($L_s$). To prove this point, we additionally leverage $L_s$, $L_{com}$ and $L_{coh}$ to train our model. The results turn out that the advanced CCNet achieves a remarkably lower style loss ($L_s$) and CCLoss ($L_{com}$ and $L_{coh}$) than all other methods, being consistent with the discussions in Sec. 4. We also report the error bars of these metrics in the Fig. 21 of the appendix.
>
> In terms of user study, we randomly select 25 content images and 30 style images, and make their complete combination to synthesize 750 stylized images for each method. For each participant, 10 stylized images of CCNet and one of the other methods are displayed in a random order. Thus, each participant needs to make 90 votes for 9 baselines and we receive 3150 response in total. During user study, the participants were asked to choose the image that learns the most characteristics from the style image. Specifically, the participants were informed that the preservation of significant style patterns was the primary evaluation point [2, 7]. Additionally, the assessment time is longer than 30 seconds for each question so that each participant can make careful decision. The Fig. 21 of the appendix illustrates the corresponding error bar. The newly added descriptions about user study are incorporated into the revision.
>
> Lastly, we will release our used test set and codes after publication to fully enable reproducibility.
>
> **[The definition / support for the $4^{th}$ claim]**
>
> Thank you for pointing this out. Originally, we aim to emphasize that we achieve better trade-off between the speed, generalization capability and stylization results over existing methods. We have dropped this claim in the revision.

---

> > ### Author Response · Authors · 2022-08-02
> > **Response to reviewer aHVM - part2**
> >
> > **[The importance of attention modules]**
> >
> > We agree that this is a very interesting point, which needs more careful exploration in the future. Especially, how to apply Transformer to realize our motivation is a very promising topic, which will become our future work. However, at the same time, some of our previous (but not restrict) results (by replacing the joint analysis module with a self-attention layer to further propagate completeness and coherence features alone) indicate that slightly increasing the size of attention modules cannot greatly advance the stylization results. We assume that, it is because the input data varies a lot in arbitrary style transfer. Therefore, compared with the data variety, the trainable parameter size of an attention module does not have critical impact on the transfer results. In contrast, different operation strategies (like enabling feature compatibility learning) have greater influences. We will dive deeper into it in the future.
> >
> > **[Results about multi-scale learning and ablation studies]**
> >
> > We politely point out that we have already reported the quantitative results for the ablations in Tab. 2 in the appendix. And the Fig. 19 illustrates the visual results by relu_4, relu_5 and their combinations, driving our current framework design.
> >
> > **[Results for Gatys et al.]**
> >
> > We conduct the results for Gatys et al. using the codes in [3], which is a very active and popular public implementation, which attracts impressive attention. And the reported results are achieved with default settings, and are consistent with (or similar to) the results in other papers [1, 4, 6]. But we do agree that Gatys et al. is very sensitive to different hyper-parameters. But for a fair and easy comparison, we simply follow their default settings like other papers.
> >
> >
> > **[The possibility of using cosine distance in Eq. 2]**
> >
> > Thank you for your insights about this detailed design. The big contrasts of content-style image patches indeed unstablize the network training if we simply use the dot product to compute similarities. To overcome this problem, we perform the normalization before feeding content-style features into the attention modules like [4, 5]. Compared with cosine distance, the normalization operation also enables CCNet to make feature diffusion based on spatial structures, which is discussed in [4, 5].
> >
> > **[Other presentation issues]**
> >
> > We have reversed the order of Fig. 9 and Fig. 10. We also expand the implementation texts and add a citation for Fig. 11 in the revision.
> >
> >
> > **Reference:**
> >
> > [1]. “Universal Style Transfer via Feature Transforms”, NeurIPS 2017.
> >
> > [2].“StyTr2: Image Style Transfer with Transformers”, CVPR 2022.
> >
> > [3].  https://github.com/anishathalye/neural-style
> >
> > [4].“Avatar-Net: Multi-scale Zero-shot Style Transfer by Feature Decoration”, CVPR 2018.
> >
> > [5].“Arbitrary Style Transfer with Style-Attentional Networks”, CVPR 2019.
> >
> > [6].“Artistic Style Transfer with Internal-external Learning and Contrastive Learning”, NeurIPS 2021.
> >
> > [7]. “Domain Enhanced Arbitrary Image Style Transfer via Contrastive Learning”, SIGGRAPH 2022.

---

> > ### Comment · Reviewer_aHVM · 2022-08-26
> > **Thanks**
> >
> > Thanks for providing more details. However, there are still quite a few question marks that I suggest addressing. However, I do not want these issues to block acceptance if the authors feel like they're not important:
> >
> >  - For the error bars in table 1 it would be good to have confidence intervals on the losses L_* to get a feeling for whether the differences between methods are significant or rather an effect of the sample size of the test set
> >  - It is not quite clear how you computed the SD for the user study. The samples are not independent, since it's a nested design with multiple users rating multiple image pairs. For proper statistics you would want an estimate of inter-observer variability to estimate the confidence interval of the scores. With the current description it's neither clear whether the error is estimated appropriately nor whether any differences between methods are significant.
> >  - The updated description of Fig. 11 does not make it more clear to me what is happening.

---

> > > ### Author Response · Authors · 2022-08-29
> > > **Thanks for your further feedback**
> > >
> > > Thank you for your further feedback. We attach the new experimental results and corresponding discussions in the uploaded revision. We also make the following explanations for better clarifications and are open to more discussions.
> > >
> > > [1]. In Tab. 3-4 of the appendix, we report the confidence intervals for the three metrics ($L_s$, $L_{com}$ and $L_{coh}$). Together with Fig. 21, we can see that the confidence intervals are quite narrow, clearly indicating the data stability of computed metrics and consistently proving our empirical superiority.
> > >
> > > [2]. Originally, we compute the standard deviation over the votes for each baseline. For example, for one baseline (e.g. Gatys et al.), we receive 350 votes to indicate whether our CCNet is superior or not in terms of the preservation of significant style patterns. And we denote 1 as a vote agreeing with our advantages, and similarly 0 as a vote preferring the baseline. Then we compute the mean and standard deviation of each baseline poll (350 votes) and report the results in Fig. 21 (d).
> > >
> > > Additionally, the mentioned sample inter-dependence problem can be addressed (or partially alleviated) by the randomly displaying strategy. Specifically, during the human evaluation for one participant, the results from baselines and our CCNet are randomly presented. For example, for $i$-th test of one subject, our result is shown on the left while the baseline result is placed at the same position for the $(i+1)$-th test. Similarly, for a single participant, the $i$-th test is designed for one baseline (e.g. Gatys et al.) but the $(i+1)$-th test corresponds to another one (e.g. WCT). The random demonstration strategy can at least partially alleviate the negative influences introduced by the sample inter-dependence problem.
> > >
> > > However, we also estimate the confidence intervals of user study by measuring the inter-observer variability. Instead of computing the mean and standard deviation of each baseline poll (350 votes), we evaluate the statistics for the feedbacks from each participant (90 votes). The results are reported in the newly updated Fig. 21 (e) with high data stability, corresponding to our advantages in human evaluation study.
> > >
> > > [3]. We have further detailed the pipeline for spatial control in the end of Sec. 4. The basic idea is to apply masks to extract specific content regions, do the transfer with corresponding style images, compute restricted stylization with mask-out operation and combine all results by adding all partial transfer results up.
> > >
> > > We hope our explanations can address your questions.

---

> > > > ### Comment · Reviewer_aHVM · 2022-08-29
> > > > **Thanks, that helps**
> > > >
> > > > [1] Just to make sure: You're showing 95% confidence intervals? It's not specified anywhere.
> > > >
> > > > [2] Ok great. Looks like a clear result indeed – at least w.r.t. your method vs. others. The other methods amongst each other is less clear from your analysis, but I guess that wasn't the objective. You could derive statements about other methods using a generalized linear mixed model and treating both images and subjects as random factors and then testing for the error over subjects. That would be interesting bonus material, but of course not needed to substantiate your claim.
> > > >
> > > > [3] So you're reassembling the masked results at the pixel level after generating multiple stylized images. I would assume that generates visible edges at the mask boundaries. Couldn't they be avoided by combining the different styles and regions at an intermediate stage, e.g. before feeding into the decoder? Again, bonus material. I believe I understand what's happening from the updated description.

---

### Author Response · Authors · 2022-08-02
**[Revision upload]**

We thank all of the reviewers for their time and effort in providing these helpful suggestions. Based on these reviews, we have raised our paper and uploaded the new version. The revised texts are highlighted in red. For each reviewer, we provide a detailed, more specific response to their points, including our changes to the revised paper.

To summarize, we have made the following changes to our paper:

**[Presentation issues]**

[1]. Adding a sub-figure in Fig. 1 to exemplify the completeness and coherence concepts.

[2]. Adding references to Fig. 5 in the introduction when talking about bi-directional softmax operation.

[3]. Removing the claim that "we achieve an optimal balance between running speed, generalization capability and stylization quality" in the introduction.

[4]. Clarifying the functionality of the normalization operation for $F_c$ and $F_s$ in Sec. 3.2.

[5]. Adding explanation for $E^i$ of Eq. (10) in Sec. 3.3.

[6]. Revising the caption of Fig. 7 to highlight the comparisons with state-of-the-art methods.

[7]. Adding more descriptions about the spatially controlled stylization in Sec. 4.


**[The quantitative experiments]**

[1]. Providing more descriptions about the conducted quantitative experiments, including the image selection strategies and user study instructions, in Sec. 4.

[2]. To more strictly follow the former papers, increasing the test set and reporting the new results in Tab. 1 and the Quantitative comparison part of Sec. 4.

[3]. To more strictly follow existing methods, introducing Deception score to further measure how the synthesized images are akin to the human-created artistic images and listing results in the $5^{th}$ row in Tab. 1.

[4]. Adding one figure to illustrate the age and gender distributions of participants in Fig. 20.

[5]. Adding one figure to illustrate the error bars for the used metrics in Fig. 21.

---

### Author Response · Authors · 2022-08-29
**[A new version uploaded]**

Dear reviewers,

Thank you for your further suggestions to improve our work. To summarize, we have made the following changes (highlighted in red) in our revision:

[1]. Adding the result of our proposed CCNet to Fig. (a) to highlight our motivation.

[2]. Adding the confidence intervals for $L_s$, $L_{com}$ and $L_{coh}$ in Tab. 3-4.

[3]. Update Fig. 21 to reflect the confidence intervals of user study.

[4]. Revising the description of Fig. 11 by adding more implementation details.

---

### Decision · Action_Editors · 2022-08-29

**Recommendation:** Accept as is

**Comment:**

The submission explicitly formalises two different objectives of image style transfer: coherence and completeness, which are the concepts that were not directly formulated in the prior works. Based on this formalisation,  the authors introduce the DNN architecture specifically tailored for higher coherence/completeness as well as the loss function that explicitly enforces better coherence/completeness.

The advantage of the proposed style transfer method is demonstrated via the results of the user study and a number of qualitative examples.

Three reviewers unanimously recommended acceptance with only minor concerns on the paper organisation, need of clarifications, and insufficient description of the user study. These concerns were successfully addressed in the revision.